



# Quantification of dynamic soil – vegetation feedbacks following an isotopically labelled precipitation pulse

Arndt Piayda*,[1], Maren Dubbert*,[2], Rolf Siegwolf[3], Matthias Cuntz[4], Christiane Werner[2]

*equal contribution

[1]Thünen Institute of Climate-Smart Agriculture, Braunschweig, 38116, Germany
[2]Ecosystem Physiology, University Freiburg, Freiburg, 79110, Germany
[3]Lab for Atmospheric Chemistry, Ecosystems and Stable Isotope Research, Paul Scherrer Institut, Villingen PSI, 5232,
Switzerland
[4]UMR Ecologie et Ecophysiologie Forestières, UMR1137, INRA-Université de Lorrain, Champenoux-54500 Vandoeuvre
Les Nancy, 54280, France

*Correspondence to*: Arndt Piayda (arndt.piayda@thuenen.de), Maren Dubbert (maren.dubbert@cep.uni-freiburg.de)

**Abstract.** The presence of vegetation alters hydrological cycles of ecosystems. Complex plant – soil interactions govern the
fate of precipitation input and water transitions through ecosystem compartments. Disentangling these interactions is a major
challenge in the field of ecohydrology and pivotal foundation for understanding the carbon cycle of semi – arid ecosystems.
Stable water isotopes can be used in this context as tracer to quantify water movement through soil – vegetation –
atmosphere interfaces.

The aim of this study is to disentangle vegetation effects on soil water infiltration and distribution as well as dynamics of soil
evaporation and grassland water – use in a Mediterranean cork – oak woodland during dry conditions. An irrigation
experiment using $\delta^{18}O$ labeled water was carried out in order to quantify distinct effects of tree and herbaceous vegetation on
infiltration and distribution of event water in the soil profile. Dynamic responses of soil and herbaceous vegetation fluxes to
precipitation regarding event water – use, water uptake depth plasticity and contribution to ecosystem evapotranspiration
were quantified.

Total water loss to the atmosphere from bare soil was as high as from vegetated soil, utilizing large amounts of unproductive
water loss for biomass production, carbon sequestration and nitrogen fixation. During the experiment no adjustments of main
root water uptake depth to changes of water availability could be observed, rendering light to medium precipitation events
under dry conditions useless. This forces understory plants to compete with adjacent trees for soil water in deeper soil layers.
Thus understory plants are faster subject to chronic drought, leading to premature senescence at the onset of drought. Despite
this water competition, the presence of Cork oak trees fosters infiltration to large degrees. That reduces drought stress,
caused by evapotranspiration, due to favourable micro climatic conditions under tree crown shading. This study highlights
complex soil – plant – atmosphere and inter – species interactions in both space and time controlling the fate of rain pulse
transitions through a typical Mediterranean savannah ecosystem, disentangled by the use of stable water isotopes.



# 1 Introduction

Vegetation influences ecosystem water cycling in many ways. Rainfall is intercepted while at the same time infiltration, redistribution and translatory flow might be altered depending on rooting depths and soil structure (Bhark and Small, 2003; Dawson, 1993; Devitt and Smith, 2002; Dubbert et al., 2014c; Schwinning and Ehleringer, 2001; Tromble, 1988). E.g., a dense vegetation layer can strongly reduce soil evaporation (Dubbert et al., 2014c; Wang et al., 2012). In turn, plant transpiration is controlled by soil water availability and distribution and plant species have different abilities to use different soil water pools (i.e. surface vs. deep or ground water). Thus, large parts of ecosystem water losses by transpiration strongly depend on plant functional types, stomatal regulation and leaf area index (*LAI*). Although studies within the last decades emphasized the pivotal role of plant roots for soil water redistribution or the role of plant transpiration on ecosystem water losses (Caldwell, 1987), it remains a major challenge to quantify dynamic soil – vegetation – atmosphere feedbacks within the water cycle.

Stable water isotopes are widely used to trace water transfers in soils, through plants and at the soil – vegetation – atmosphere interface (Werner and Dubbert, 2016; Yakir and Sternberg, 2000). Fractionation between the heavier and lighter isotopes occurs during phase changes (from liquid to gaseous, equilibrium fractionation) and movement (kinetic fractionation). This leads to different stable isotope compositions ($\delta^2H$ and $\delta^{18}O$) in various water pools (i.e. rain, groundwater), along soil profiles, in different plant species and between water vapor evaporated from soil compared to water transpired by plants. These differences provide the basis for tracing water through an ecosystem. For example, utilization of different water pools within the soil by different plant individuals may be possible (Dawson, 1993; Volkmann et al., 2016a). Isotopes can further help to separate transpiration from soil evaporative fluxes (Dubbert et al., 2013; Yepez et al., 2003) or to study infiltration or distribution of precipitation in soils (Garvelmann et al., 2012; Rothfuss et al., 2015). Stable water isotopes have also been used to study water movement at the soil – vegetation interface (Caldwell et al., 1998). The isotopic composition of plant water uptake can be determined by sampling the 'output' of the root system, for example the plant xylem, because the water isotopic signatures are usually not altered by plant water uptake (Dawson, 1993). Compared with values observed in the soil water profile, the preferential plant extraction depth or the proportional use of "event water" (i.e. singular precipitation events) can be determined. Although this method has been successfully used to identify processes such as hydraulic lift and soil water redistribution (Caldwell et al., 1998), most datasets were limited in temporal and spatial resolution. Over the last decade, the development of field – deployable laser spectroscopy has enabled continuous measurements of water vapour and its isotopic signatures in ecosystem fluxes and atmospheric concentrations. This opens the door for large-scale assessment of the soil-vegetation-atmosphere interactions in the water cycle. In particular, these developments have enhanced the spatial and temporal resolution tremendously, furthering the understanding in the fields of plant ecophysiology (Cernusak et al., 2016) and ecosystem physiology (Dubbert et al., 2014a; Dubbert et al., 2014c).

In the present study, we focus on disentangling the vegetation effects on soil water infiltration and distribution as well as dynamics of soil evaporative and grassland water-use in a Mediterranean cork-oak woodland. An irrigation experiment with



$\delta^{18}O$ labeled water was carried out to quantify the distinct effects of tree and herbaceous vegetation on 1) infiltration and distribution of "event water" (freshly introduced water) in the soil profile and 2) to quantify the dynamic responses of soil and herbaceous vegetation fluxes to precipitation regarding event water-use, plasticity of water uptake depth and contribution to ecosystem *ET*.

# 2 Material and methods

## 2.1 Study site and experimental design

Measurements were conducted between May 26 and June 6 2012 in an open cork-oak woodland (*Quercus suber* L.) in central Portugal, approximately 100 km north-east of Lisbon (N39°8'17.84'' W8°20'3.76''; Herdade de Machoqueira do Grou). The trees are widely spaced (209 individuals ha$^{-1}$) with a leaf area index of 1.1 and a gap probability of 0.7 (Piayda et
al., 2015).

The herbaceous layer is dominated by native annual forbs and grasses. The site is characterized by Mediterranean climate, with a 30 year long-term mean annual temperature of approximately 15.9 °C and annual precipitation of 680 mm (Instituto de Meteorologia, Lisbon). We established two sites: one directly under the oak crown projected area (tree site, ts) and another one in an adjacent open area (open site, os). Two types of plots (sized 40 × 80 cm) were installed in each site: bare
soil plots with total exclusion of above and below – ground biomass (root in – growth was prevented by inserting trenching meshes, mesh diameter < 1 μm, Plastok, Birkenhead, UK), and understory plots with herbaceous vegetation (four plots per site and treatment). All plots were established 1 year before measurements to minimize effects of disturbance (For further details see Dubbert et al. (2013)).

After a base line observation, all plots were watered with 20 mm water of an oxygen isotopic signature of -139.5‰ to trace
the influence of different vegetation components on water infiltration. Thereafter, all measurements were conducted in 7 diurnal cycles over the following 10-12 days. The open and tree sites were watered independently, as the measurement setup did not allow highly resolved observations of all treatment plots at the same time. Environmental variables (PPFD; soil water content; vpd) were not significantly different between the first and second half of the observation period.

## 2.2 Environmental variables and plant parameters

Photosynthetic photon flux density (PPFD) was measured at both sites at approximately 1.5 m height (PPFD, LI-190SB, LI-COR, Lincoln, USA). Rainfall (ARG100 Rain gauge, Campbell Scientific, Logan, UT, USA), air temperature, and relative humidity (rH, CS-215 Temperature and Relative Humidity Probe, Campbell Scientific, Logan, UT, USA) were measured and 30 min averages were stored in a data logger (CR10x, Campbell Scientific, Logan, UT, USA). Soil temperature (custom built pt-100 elements) in -5 cm depth was measured in vegetation and bare soil plots at both sites and 60 min averages were
stored in a data logger (CR1000, Campbell Scientific, Logan, UT, USA; 4 sensors per depth and treatment). Temperature at the soil surface was manually measured on each measurement day in diurnal cycles corresponding with the gas exchange





measurements using temperature probes (GMH 2000, Greisinger electronic, Regenstauf, Germany). Volumetric soil water content ($\theta_s$, 10hs, Decagon, Washington, USA) in -5, -15, -30, and -60 cm depth was measured on vegetation and bare soil plots at both sites and 60 min averages were stored in a data logger (CR1000, Campbell Scientific, Logan, UT, USA; 4 sensors per depth and treatment).

Living aboveground biomass of herbaceous plants was determined destructively on five randomly selected, $40 \times 40$ cm plots at the beginning and end of the experiment in the open and under the trees. All green fresh aboveground plant biomass was collected, divided by species, dried (60 °C, 48 hours) and weighed. Total aboveground biomass was relatively low compared to previous years between 42 and 78 g m-2 (see Fig. A1), due to the considerable winter/spring drought in the hydrological year 2012 (Costa e Silva et al., 2015; Dubbert et al., 2014b; Piayda et al., 2014). While total aboveground biomass was

similar between plots, species composition and relative dominance differed with the open sites being dominated by *Tuberaria guttata* and the tree sites by grass and legume species (Dubbert et al., 2014b).

**2.3 Cavity Ring-Down Spectrometer based gas-exchange flux and δ¹⁸O measurements**

Water fluxes and isotopic composition were measured with a Cavity Ring-Down Spectrometer (CRDS, Picarro, Santa Clara, USA) in combination with custom built soil chambers (following the design of Pape et al. (2009)) in an open gas exchange

system (n=3 per treatment and experimental site). Background and sampling air were measured alternately after stable values were reached. A five minutes interval average was used for the calculation of evapotranspiration (*ET*) and evaporation (*E*). *ET, E* as well as total conductance ($g_t$) were calculated according to von Caemmerer and Farquhar (1981). Oxygen isotope compositions of soil evaporation (bare soil plots) as well as evapotranspiration of the understory (vegetation plots) were estimated using a mass balance approach (Dubbert et al. (2013); Dubbert et al. (2014c)):

$$\delta_E = \frac{u_{out}w_{out}\delta_{out} - u_{in}w_{in}\delta_{in}}{u_{out}w_{out} - u_{in}w_{in}}$$

$$= \frac{w_{out}\delta_{out} - w_{in}\delta_{in}}{w_{out} - w_{in}} - \frac{w_{in}w_{out}(\delta_{out} - \delta_{in})}{w_{out} - w_{in}}$$

(1),

where $u$ is the flow rate [mol(air) s$^{-1}$], $w$ is the mole fraction [mol(H$_2$O) mol(air)$^{-1}$] and $\delta$ is isotope value of the incoming (*in*) and outgoing (*out*) air stream of the chamber. Flow rates are measured with humid air so that conservation of dry air gives $u_{in}(1-w_{in}) = u_{out}(1-w_{out})$, which leads to the second line of Eq. (1). The second term in Eq. (1) corrects for the increased gas flow in the chamber due to addition of water by transpiration. In addition to isotopic signatures of soil evaporation and understory evapotranspiration, the oxygen isotope signatures of ambient water vapor (in 9 m height) were measured with the

CRDS. All measurements were conducted as diurnal courses with 5-6 measurement points between 7 a.m. and 7 p.m. For more details about the chamber design and measurement setup see Dubbert et al. (2013).





## 2.4 Sampling and measurement of $\delta^{18}O$ of soil and leaf water

Soil samples for water extraction and $\delta^{18}O$ analysis were taken on vegetated and bare soil plots using a soil corer. Samples were collected from the soil surface (0-0.5 cm depth), -2, -5, -10, -15, -20, and -40 cm soil depths (n=4 per depth and treatment) usually during midday, but on the day of irrigation directly proceeding the irrigation pulse and additionally at

18:00. Mixed leaf samples of the herbaceous vegetation for water extraction were obtained in daily cycles in 2-hourly steps from 8:00 to 18:00. Soil and leaf water samples were extracted on a custom build vacuum line by cryogenic distillation. Water $\delta^{18}O$ analysis was performed by headspace equilibration on an Isoprime IRMS (Elementar, Hanau, Germany) coupled via open split connection to a μgas autosampler (Elementar, Hanau, Germany). Equilibration with 5% He gas was done for 24 hours at 20 °C. For every batch of 44 samples 3 different laboratory standards were analyzed. Laboratory standards were

regularly calibrated against VSMOW, SLAP, and GISP water standards (IAEA, Vienna). Analytical precision was 0.1‰.

## 2.5 Partitioning of evapotranspiration

Oxygen isotope signatures of soil evaporation were calculated using the equation of Craig and Gordon (1965):

$$R_E = \frac{1}{\alpha_k \alpha^+ (1-h)} (R_e - \alpha^+ h R_a) \tag{2},$$

where $R_E$ is the isotope ratio ($^{18}O/^{16}O$) of evaporated water vapor and $R_e$ is the isotope ratio of bulk soil water at the

evaporating sites. The evaporating site is the vapor-liquid interface below which liquid transport and above which vapor transport is dominant (Braud et al., 2005). It has been shown for unsaturated soils that this site is related to a strong enrichment in soil water isotopic composition relative to the rest of the soil column and an exponential depletion in isotopic signature within few cm of the underlying soil due to evaporative enrichment of the remaining liquid water (Dubbert et al., 2013; Haverd and Cuntz, 2010). Thus, for $R_e$ and temperature at the evaporating sites ($T_e$), temperature and oxygen isotope

signatures of bulk soil water were measured along the soil profile and those values along the soil profile were used where the strongest enrichment in bulk soil $\delta^{18}O$ could be detected (residual soil water volumetric content was only 1% and therefore neglected). $R_a$ is the isotope ratio of ambient water vapor, $\alpha_k$ is the kinetic fractionation factor, $\alpha^+$ is the water vapor equilibrium fractionation factor ($\alpha_k$ and $\alpha^+ > 1$; Majoube (1971); Merlivat (1978); for the formulation of $\alpha_{k=} \alpha_{diff}^{nk}$ (Mathieu and Bariac, 1996)), and $h$ is the relative humidity normalized to $T_e$.

Although direct estimates of $E$ and $\delta^{18}O_E$ were available for bare soil plots, vegetation depresses $E$ and also influences $\delta^{18}O_E$, for example due to different isotopic signatures of soil water and also temperature at bare soil and vegetated soil patches (Dubbert et al., 2013). Therefore, bare soil plots only served to validate the Craig and Gordon equation, because on bare soil plots $E$ contributes entirely to the evaporative flux and could be tested against modeling results. Finally, the Craig and Gordon equation was used to calculate $\delta^{18}O_E$ of vegetation plots.

The oxygen isotope signature of transpired water vapor $\delta^{18}O_T$ was calculated based on the isotopic signature of bulk leaf water $\delta^{18}O_L$ using the Craig and Gordon equation (Eq. 2) instead of measuring xylem/source water isotopic signatures and





modeling $\delta^{18}O_L$ of leaf water at the evaporating sites due to the lack of suberized/lignified plant parts in the herbaceous vegetation. The isotopic signature on the evaporating site $\delta^{18}O_e$ was thus estimated by:

$$\delta^{18}O_e = \frac{\delta^{18}O_L \wp}{1 - e^{-\wp}} \text{ with the Péclet number } \wp = \frac{TL_{eff}}{CD} \tag{3},$$

where $L_{eff}$ is the effective path length of water movement in the leaf mesophyll (0.05 m), $C$ is the molar water concentration (55.6 × 10³ mol m⁻³) and $D$ is the tracer diffusivity in liquid water (2.66 × 10⁻⁹ m² s⁻¹). $T$ was estimated iteratively with equation (Eq. 4) using $ET$ as initial value. Convergence was generally reached after five iterations. Small differences in isotopic compositions were found compared to a direct use of $\delta^{18}O_L$ in equation (Eq. 2), which were not significant for results shown throughout this work.

Finally, the contribution of $T$ to $ET$, $ft = T/ET$, can be estimated based on measured understory $\delta^{18}O_{ET}$ and modeled soil $\delta^{18}O_E$ and herbaceous $\delta^{18}O_T$ (Moreira et al. (1997);Yakir and Sternberg (2000)):

$$ft = \frac{\delta^{18}O_{ET} - \delta^{18}O_E}{\delta^{18}O_T - \delta^{18}O_E} \tag{4}.$$

This approach is based on the assumption that the isotopic signature of evapotranspiration is a mixing ratio of not more than the two sources (evaporation and transpiration) and that no water vapor is lost other than by the mixing of the two sources with the atmospheric pool (i.e. no condensation).

## 2.6 Event water partitioning

Event water describes the amount of water in ecosystem pools or fluxes that originates from a certain rain event. To calculate the amount of event water in volumetric soil water content $\theta$ that originates from the isotopically labeled watering event, the following two-source mixing model was used:

$$f_{\theta,eve} = \frac{\delta^{18}O_\theta - \delta^{18}O_{\theta,pre}}{\delta^{18}O_{eve} - \delta^{18}O_{\theta,pre}} \tag{5},$$

where $f_{\theta,eve}$ is the fraction of rain event water in $\theta$ at a certain time after the event, $\delta^{18}O_\theta$ is the stable isotope ratio in $\theta$ at a certain time after the event, $\delta^{18}O_{\theta,pre}$ is the stable isotope ratio of soil water before the rain event and $\delta^{18}O_{eve}$ is the stable isotope ratio of the precipitation event water. The model assumes no fractionation of rain event water during infiltration and was solved separately for each depth. Contributions of infiltrated event water to evaporation fluxes from soil and transpiration fluxes from plant surfaces were calculated analogously:

$$f_{E,eve} = \frac{\delta^{18}O_E - \delta^{18}O_{E,pre}}{\delta^{18}O_{E,eve} - \delta^{18}O_{E,pre}} \tag{6},$$

$$f_{T,eve} = \frac{\delta^{18}O_T - \delta^{18}O_{T,pre}}{\delta^{18}O_{T,eve} - \delta^{18}O_{T,pre}} \tag{7},$$

where $f_{E,eve}$ and $f_{T,eve}$ are the fractions of rain event water in evaporation $E$ and transpiration $T$. $\delta^{18}O_{E,pre}$ and $\delta^{18}O_{E,eve}$ are the isotopic compositions of evaporation calculated with equation (Eq. 2) assuming that the source water isotopic composition equals either $\delta^{18}O_{\theta,pre}$ at the evaporative site or $\delta^{18}O_{eve}$, respectively. $\delta^{18}O_{T,pre}$ and $\delta^{18}O_{T,eve}$ are the isotopic compositions of



transpiration calculated with equation (Eq. 2) and (Eq. 3) assuming that the source water isotopic composition equals either bulk leaf composition before watering $\delta^{18}O_{L,pre}$ or $\delta^{18}O_{eve}$, respectively.

## 2.7 Root water uptake

The allocation of root water uptake by plants along the soil depth was estimated via a three-source model. Therefore, the isotopic composition of transpiration $\delta^{18}O_T$ calculated with equation (Eq. 2 and 3) from three independent observations of leaf water compositions $\delta^{18}O_L$ were compared with three independent solutions for isotopic transpiration composition $\delta^{18}O_T$ of equation (Eq. 2), each assuming the current water source for transpiration originating only from an observed depth ($d1 = -5$ cm, $d2 = -15$ cm, $d3 = -30$ cm). Soil depths above and below $d1$ to $d3$ showed negligible root density in the profile and could therefore be excluded from the model. The three possible source fluxes are related to the resulting transpiration flux mixture via the following system of equations (compare e.g Philips et al. (2005)):

$$\delta^{18}O_{T1} = f_{T,d1} \cdot \delta^{18}O_{T1,d1} + f_{T,d2} \cdot \delta^{18}O_{T1,d2} + f_{T,d3} \cdot \delta^{18}O_{T1,d3} + \varepsilon_1$$

$$\delta^{18}O_{T2} = f_{T,d1} \cdot \delta^{18}O_{T2,d1} + f_{T,d2} \cdot \delta^{18}O_{T2,d2} + f_{T,d3} \cdot \delta^{18}O_{T2,d3} + \varepsilon_2$$

$$\delta^{18}O_{T3} = f_{T,d1} \cdot \delta^{18}O_{T3,d1} + f_{T,d2} \cdot \delta^{18}O_{T3,d2} + f_{T,d3} \cdot \delta^{18}O_{T3,d3} + \varepsilon_3$$ (8),

$$1 = f_{T,d1} + f_{T,d2} + f_{T,d3}$$

where $f_{T,d}$ denotes the fraction of source water contribution from depths $d1$ to $d3$ to the transpiration flux. The system was solved for $f_{T,d1}$ to $f_{T,d3}$ using a shuffled complex evolution algorithm (Duan et al., 1992) minimizing a multi-objective cost function (Duckstein, 1981) combining the error terms $\varepsilon_1$ to $\varepsilon_3$ for each time step.

## 3 Results

### 3.1 Environmental and soil conditions

Tree cover significantly influenced diurnal courses of incoming global radiation $R_g$ during daytime on the sites. Strong reductions of $R_g$ between 09:00 and 18:00 o'clock reduced daily sum of energy input $\sum R_g$ by 17.1 MJ m$^{-2}$d$^{-1}$ on the open sites (os) compared to the tree sites (ts) (Fig. 1). However, air temperature and relative humidity was very similar in the open area and below trees with mean values around 66% and 19°C throughout the experiment. Similar to $R_g$, the amplitude of daily mean soil temperatures $T_S$ in the upper soil layer were smaller on tree sites (bare: 7.4 °C, veg: 5.5 °C) than in the open area (bare: 14.9 °C, veg: 11.3 °C, Fig. 1). In contrast, understory vegetation cover reduced the soil temperature only by 2-3.6 °C on both sites.

Soil moisture $\theta$ prior to the irrigation pulse ranged from 5 – 10% (Fig. 3), which is low compared to the annual average, but typical for the observation period at the end of May and the beginning of the dry season. Systematically lower soil moisture



$\theta$ at depths below 20 cm could be observed at the tree sites located close to trees compared to open sites, whereas the upper soil layers showed comparable values for all sites prior to the experiment.

### 3.2 Oxygen isotope signatures of ecosystem water pools

Stable oxygen isotope composition of soil water $\delta^{18}O_S$ for all plots and all depths ranged between -7.3‰ and 10.1‰ before the irrigation. Compared to the very depleted irrigation water signature of -139.5‰, only small enrichment in $\delta^{18}O_S$ of on average 0.4‰ in the open sites compared to the tree sites were found and 2.9‰ enrichment of bare soil compared to vegetation plots preliminary to the watering (Fig. 2). Irrigation caused a strong depletion of $\delta^{18}O_S$ with a peak only 1 h after irrigation in the upper soil layer. Lowest $\delta^{18}O_S$ values were found at tree sites on bare soil plots with $\delta^{18}O_S$ = -106.06‰ and tree sites with vegetation cover with $\delta^{18}O_S$ = -85.1‰ whereas the open sites showed weaker maximum depletions of $\delta^{18}O_S$ = -79.9‰ and $\delta^{18}O_S$ = -49.4‰ on bare soil and vegetation plots, respectively. The nine days following the irrigation event were characterized by a steady increase of $\delta^{18}O_S$, which was only slightly depleted compared to pre-event $\delta^{18}O_S$ nine days after irrigation. In addition to the absolute differences in peak $\delta^{18}O_S$ between sites, the depletion in $\delta^{18}O_S$ was maintained for a longer period at tree sites (Fig. 2).

Oxygen isotope signatures of soil evaporation and leaf water as well as transpired water vapour (Fig. 4) showed an immediate response to the irrigation pulse with peak depletion only 1 hour after labelling for soil evaporation and 3 hours for leaf water and transpired vapour. Subsequently, an exponential rise to pre – event isotope values could be observed in all pools. Depletion in $\delta^{18}O_E$ of soil evaporation was much stronger compared to $\delta^{18}O_T$ of plant transpiration (and leaf water $\delta^{18}O_L$). $\delta^{18}O_E$ of soil evaporation and evapotranspiration $\delta^{18}O_{ET}$ were both significantly more reduced on the tree sites compared to the open sites. A similarly strong vegetation effect could be seen between $\delta^{18}O_E$ on bare soil plots in comparison to understory vegetation plots.

### 3.3 Infiltration and distribution of event water

Daily mean soil moistures $\theta$ throughout the experiment were characterized by the ongoing drought at all sites (Fig. 3). Watering the plots with 20 mm increased mean daily soil moisture $\theta$ in the upper layers only by 2%$_{vol.}$ to 6%$_{vol.}$ and had no effect on deeper soil layers. However, partitioning event water fractions revealed an extensive replacement of old, pre-event water with new event water up to 4%$_{vol.}$ and even down to depths below -30 cm (Fig. 3), in particular on bare soil plots. Systematically increased infiltration and deepened distribution of event water was observed on tree sites compared to open sites. In the course of the experiment, soil moistures returned to pre-event values and below. The decrease of event water was here much stronger than of pre-event water, leaving nearly no trace nine days after the watering.

### 3.4 Event water use by soil evaporation and plant transpiration

While pre-event $E$ on bare soil plots was lower than $ET$ on vegetation plots on both the open and tree sites, $E$ and $ET$ equally peaked with roughly 3.3 mmol m$^{-2}$ d$^{-1}$ on the open sites. However, on the tree sites post-event peak of $E$ at bare soil plots (2.1



± 0.1 mmol m$^{-2}$ d$^{-1}$) was higher than $ET$ at vegetation plots (1.5 ± 0.2 mmol m$^{-2}$ d$^{-1}$). Moreover, the peak of $ET$ on both sites was shifted by 24 h compared to $E$ and occurred only 2 days after irrigation (Fig. 5). Following peaks in $E$ and $ET$, evapotranspiration losses declined exponentially to pre-event values 3 days after irrigation on all sites.

Partitioning $ET$ on vegetation plots on both sites into soil $E$ and plant transpiration $T$ revealed that the time shift of the response of the $ET$ flux compared to bare soil plots $E$ was caused solely by a slower reaction of $T$ to the irrigation pulse. Throughout the experiment the proportion of $T$ to $ET$ ranged from 9% to 59% on open sites and 17% to 66% on shaded sites. Event water fraction in soil evaporation $f_{E,eve}$ and plant transpiration $f_{T,eve}$ differed considerably with $T$ utilizing only a peak of 12% of the event water while $E$ is fed up to 62% by event water following irrigation (Fig. 6). Nine days after the irrigation pulse event water contribution of $T$ and $E$ converged on average to 10% of the respective flux and differences between $f_{E,eve}$ and $f_{T,eve}$ faded. Event water use of soil evaporation $f_{E,eve}$ showed no significant differences between open and tree sites nor between bare soil plots and vegetated plots except on the day of watering on the open vegetation plot. Here, $f_{E,eve}$ reached only about 25%, corresponding to the limited availability of event water in the soil (Fig. 2). Along the lines of evaporation, no significant differences could be observed between $f_{T,eve}$ on open and vegetation plots.

### 3.5 Root water uptake allocation

Prior to the irrigation pulse we refrained from calculations of root water uptake allocation since the differences in $\delta^{18}O_S$ along soil depth were too small (see above) for a sufficiently accurate prediction power solving equation system (Eq. 8) and derive significant $f_{T,d}$. Following the label pulse, soil water uptake by plants was located solely at soil depths around -30 cm with no change in time or between open and tree sites despite a small uptake of water for transpiration from soil layers around -15 cm on day 0 and 1 after watering (Fig. 7).

## 4 Discussion

### 4.1 Infiltration and distribution of event water

Mosaic patterns of vegetation cover by understory plants and trees are characteristic for savannah-type ecosystems (Belsky, 1994; Greig-Smith, 1979). Different vegetation cover is known to alter soil hydrological conditions and micro climate (Scholes and Archer, 1997) which in turn have effects on vegetation cover and ecosystem sustainability in future climate change scenarios (Breman and Kessler, 1999; Pueyo et al., 2012). Infiltration of event water into soil in this ecosystem is strongly altered by understory cover and tree shading. Vegetation cover of understory plants reduced infiltration on average by 24% compared to bare soil (Fig. 3). The reason can be found in interception, subject to instantaneous plant and litter surface evaporation before the first flux observations, which was took place one hour after watering. This water uptake limitation could neither be compensated by plant roots, breaking the crust formations which can be observed in the field and are common for Mediterranean soils limiting hydraulic conductivity of top soils (Eldridge et al., 2010; Goldshleger et al., 2002; Maestre et al., 2002) nor by beneficial shading effects by the above ground biomass, which did not significantly



reduce the soil surface temperatures significantly (Fig. 1) and thus, the evaporative demand of boundary layers. The observed infiltration on the day of watering can further be regarded as unaffected by understory root water uptake confirmed by low transpiration fluxes on the day of watering (Fig. 5). In contrast to previous studies, which report beneficial effects of plant cover on infiltration conducted year-round in the wet year 2011 (Dubbert et al., 2014c). This study is focused on the

transition period between spring and the onset of summer drought, during the exceptionally dry year 2012 with high atmospheric evapotranspirative demands intensifying interception losses. This unexpected turn in effect direction, which depends on plant cover and atmospheric evapotranspirative demand potentially plays a strong role for the water balance of the ecosystem in the course of the ongoing climate change scenarios (compare Fig. A1 and Dubbert et al. (2014b), max. biomass $70 \pm 21$ g m$^{-2}$ and 89 % cover in 2011 and 55 g m$^{-2}$ biomass and 38 % cover in 2012).

Tree shading had a tremendous impact on the microclimate above understory plant and soil surfaces, but effects on infiltration amount could only be observed on vegetated plots. Reductions of the daily sum of global radiation $\sum R_g$ by 72% and daily peak soil temperatures $T_{S,5cm}$ up to  22% (Fig. 1) generated favorable conditions. Limited instantaneous evaporation from plant surfaces as described above led to 71% higher infiltration amounts (Fig. 3), whereas the anyway high infiltration amounts on bare soil plots were unaffected tree shading. Previous studies reported similar, positive feedbacks of tree cover

for the hydrological cycle in savannah-type ecosystems, which were not only related to shading effects (Eldridge and Freudenberger, 2005) but to the actual change of soil hydraulic properties beneath tree crowns (Bargués Tobella et al., 2014). Supporting findings are given by (Bhark and Small, (2003) and D'Odorico and Porporato, (2006). Considering the projected shading by crown cover of the tree layer (minimum of 30% at noon, increasing during the rest of the day, Piayda et al. (2015)), the infiltration enhancement has potentially large benefits on the ecosystem level. However, the impact of canopy

interception losses in this ecosystem, potentially exceeding the infiltration benefits of tree cover (compare Joffre and Rambal (1993) and Dubbert et al. (2014c)), could not be analyzed in this study and needs further investigations with regard to tree density and age.

Subsurface distribution of soil water $\theta$ was systematically lower at depths below -20 cm at tree sites compared to open sites (Fig. 3). This clearly indicates the enhanced water extraction by tree roots close to trees, similar to results of Dubbert et al.

(2014b). The observed pattern could not be changed by the event water pulse of 20 mm, equal to a rain event of moderate intensity on this site. That explains the intense drought stress understory plants suffer during the transition period from moist spring to dry summer, leading to earlier dieback under tree cover (Dubbert et al., 2014b; Moreno, 2008). The depth distribution of event water is very similar on bare soil plots that show an over all deeper infiltration of more water than the vegetated plots, caused by the higher infiltration amounts shown before. This shortcoming could partially be compensated by

higher infiltration amounts below tree shading, but was consumed by tree water uptake from deeper depths within one day. During these dry conditions, pre-event water is located in small pores under high matrix potentials. Infiltrating event water partially displaced pre – event water downwards (Fig. 3) and additionally filled larger pores in the top soil. Thus, event water is more subject to evaporation due to lower matrix potentials in bigger pores than pre-event water. This observation is supported by a rapid decrease of event water content throughout the experiment.



## 4.2 Dynamic responses of event water-use and plasticity of water uptake depth

Successful biomass production of herbaceous vegetation highly depends on soil water availability in upper soil layers hosting the root system. Occasional precipitation events control the soil water regime (Porporato et al., 2004) which are prone to substantial changes in future climate change scenarios by stronger short term fluctuations of drought events (IPCC, 2013). Thus, a rapid adaptation of preferential root water uptake depth is crucial. This is particularly important for herbaceous vegetation in order to maximize the utilization of different soil water pools for a successful seed production, longevity and inter species competition (Ehleringer and Dawson, 1992; Rodriguez-Iturbe, 2000). It could be clearly shown that understory transpiration $T$ responded slower to an incoming precipitation pulse than soil evaporation $E$, with a time lag of about 24h. During the entire experiment, $E$ was the dominant flux on both, tree and open sites, with a comparable contribution of transpiration $T$ to evapotranspiration $ET$ of 36% and 41% (Fig. 5), respectively. This small loss of productive water originates on one hand from the longer time response lag of $T$. On the other hand that only little event water reaches deeper soil layers where understory plants have their main root water uptake depth prior to the precipitation pulse. Event water use of the understory vegetation was overall low, since no shift of root water uptake depth could be observed within the nine days of the experiments (Fig. 7) leading to comparably small isotopic depletion of bulk leaf water and transpiration (Fig. 4). This is in agreement with previous findings where annual savannah species were not fast enough readjusting their water extraction depth in order to exploit precipitation water more efficiently (Asbjornsen et al., 2008; Kulmatiski and Beard, 2013). More importantly, during that period of the year the dry conditions in the upper soil layers forces understory plants in the direct vicinity of trees to compete for soil water at lower depths where the trees have their roots (i.e. tree sites). This observation clearly opposes the widely discussed two-layer hypothesis, proposing independent ecological niches for root water uptake of trees and understory plants in savannahs in order to avoid competition (Hipondoka et al., 2003; Holdo and Planque, 2013; Kulmatiski et al., 2010; Walter et al., 1971). Quite the contrary, previous findings of, e.g. Pryardarshini et al. (2015), suggest that tree-based soil water redistribution by hydraulic lift (Dawson, 1993) is an important contribution in water limited ecosystems like savannahs. This is a possible explanation for understory root water uptake at the depth of the first tree roots, as we found it in our study. Moreover, exponential soil profiles of plant available nitrogen causes a coupled water and nutrient competition between herbs and trees in this ecosystem during spring (Dubbert et al. , 2014). Modeling studies of e.g. Nippert et al. (2015) already suggested that understory plants do not exploit all accessible soil layers (including the top layers with high drought risk) in order to maximize water availability. Lower, but more resilient production is achieved instead by limiting root growth and water uptake to deeper depths, which could be confirmed by this study.

Recently, Volkmann et al. (2016a) used a similar flux / isotope approach to test the widespread dogma that plant water uptake depth is primarily controlled by root density distribution. While grassland species did not strongly alter their uptake pattern during the measurement campaign their water uptake depth profile was not in accordance with their root density distribution, with 85 % in the upper 10 cm of the soil profile. This clearly indicates that adapting the water uptake to soil



water availability plays a role, but probably on longer time scales than what we observed during the 10 day's lasting experiment. Therefore, the development of membrane-based *in-situ* methods of soil water (Gaj et al., 2015; Rothfuss et al., 2015; Volkmann et al., 2016a) and xylem sap sampling (Volkmann et al., 2016b) will advance the studies of dynamic changes in eco – hydrological soil – vegetation feedbacks in the future. Furthermore, the coupling of isotope laser

spectroscopes to gas-exchange chambers and soil or xylem equilibration probes overcomes the cost and time consuming classical destructive sampling methods. Recent studies (i.e. Orlowski (2013)) showed significant isotopic deviations between actual soil water that is available for the plants and water that is cryogenically extracted from soil samples depending on soil the type. While we did not observe this in the sandy soils at our study site, these effects might severely hamper the usefulness of destructive soil sampling techniques in clay or loam soils. The newly developed in – situ techniques will thus

facilitate cost – effective measurements of soil or xylem isotopic signatures with highest resolution, enhancing our capacity to study the dynamics in soil water infiltration, in the uptake of water by plants and in the partitioning of evapotranspiration.

**5 Conclusion**

In this study, the various interactions between understory vegetation and trees of a Mediterranean cork – oak woodland affecting the ecosystem water flows could be disentangled and quantified. The immediate on-site determination (with

CRDS) of the isotope ratios from different soil and ecosystem compartments in combination with in – situ sampling methods enhanced the resolution, precision and reliability of our results. This facilitated the tracing of the fate of rain pulse transitions through a typical Mediterranean savannah ecosystem using stable water isotopes.

Irrespective of the presence of vegetation or just bare soil, the total evapotranspirative water loss of soil and understory remains unchanged. Thus, the amount of unproductive water loss is largely reduced, in favor of biomass production, carbon

sequestration and nitrogen fixation. Adjustments of main root water uptake depth to changing soil water availability after rain pulses could not be observed. Consequently the understory plants could not utilize light to medium precipitation. Therefore these understory plants were forced into water competition with trees, rooting at deeper soil layers. However, the understory plants could profit from tree root induced soil water redistribution. Cork oak trees foster infiltration to large degrees and considerably reduce understory and soil evapotranspiration by altered micro climatic conditions under tree

crown shading. Despite these benefits, understory plants in immediate vicinity of trees suffer from systematically lower soil moistures in deeper layers leading to premature senescence at the onset of drought. Complex soil – plant – atmosphere and inter – species interactions could be successfully disentangled in both space and time.



**Appendix A**

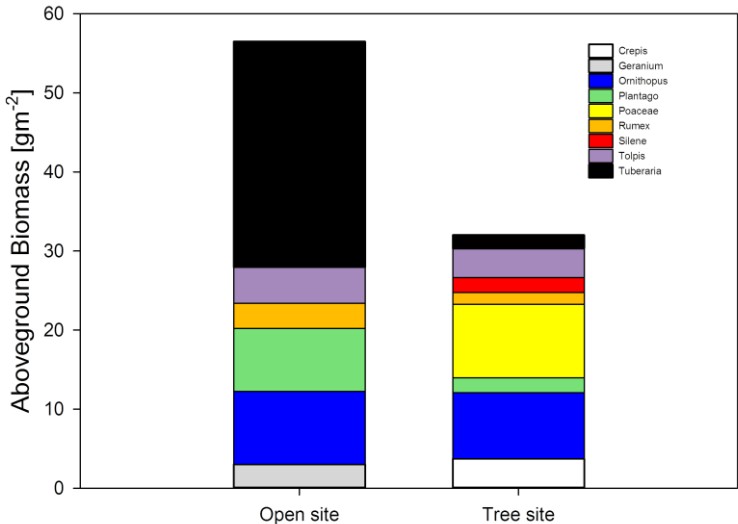

**Figure A1: Aboveground biomass on vegetated plots during the experiment time given for each genus. Standard errors are not given for the sake of clarity, but amount on average 30% of displayed genus biomass.**

**Author contribution**

Arndt Piayda and Maren Dubbert contributed equally to experimental work, data analysis and writing the manuscript. Rolf Siegwolf proofread the manuscript. Matthias Cuntz contributed to data analysis and proofread the manuscript. Christiane Werner contributed to field work and proofread the manuscript.

**Competing interests**

The authors declare that they have no conflict of interest.

**Acknowledgements**

We gratefully acknowledge excellent help in the laboratory by Ilse Thaufelder. Funding for this study was provided by the DFG (WATERFLUX Project: # WE 2681/6-1, # CU 173/2-1).

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



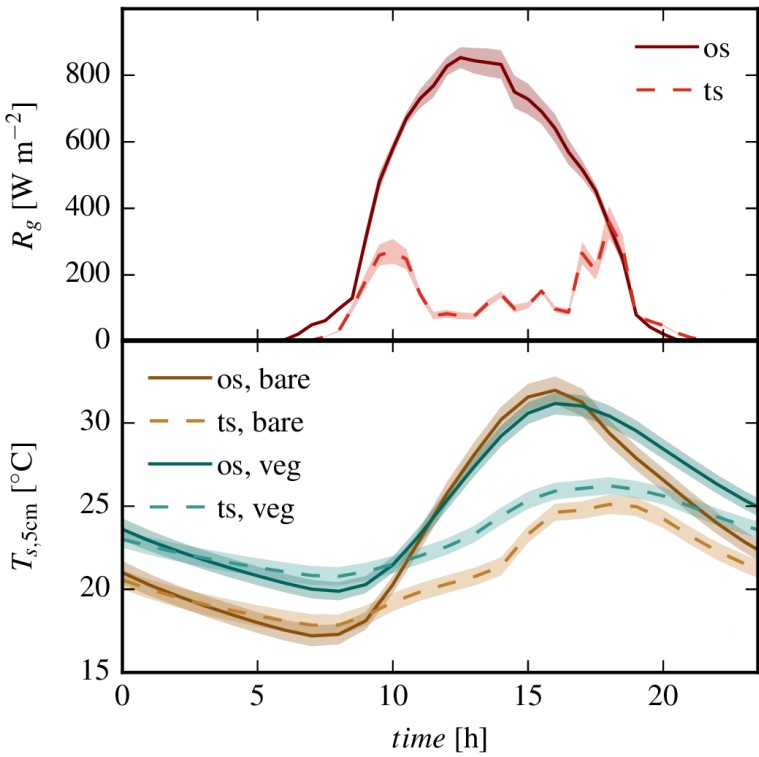

**Figure 1:** Daily cycles, averaged over the experiment period, of a) global radiation $R_g$ in 1.5 m height and b) soil temperature $T_{s,5cm}$ in -5 cm depth under bare soil (bare) or vegetation cover (veg). Observations at open sites between tree crowns (os) and shaded sites beneath tree crowns (ts) are shown. Uncertainty bands display standard error.

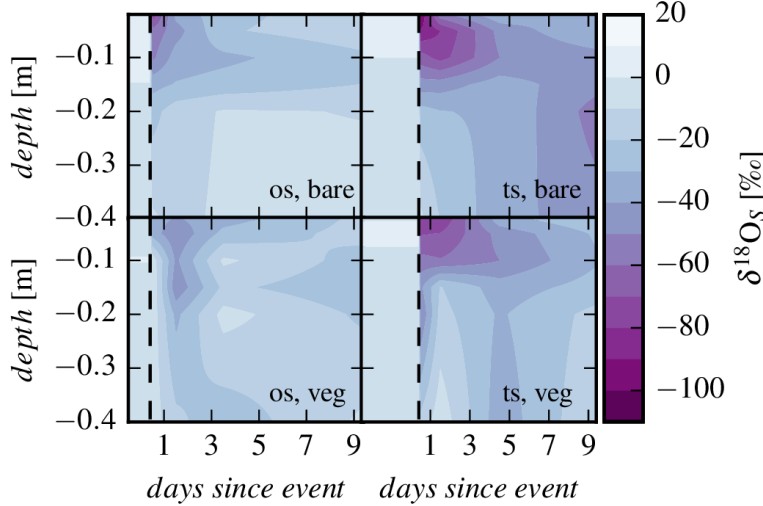

**Figure 2:** Mean daily isotopic composition of soil water $\delta^{18}O_S$ during experiment under bare soil (bare) or vegetation cover (veg) at open sites between tree crowns (os) and shaded sites beneath tree crowns (ts). Dashed lines mark time of watering event. Interpolation method: linear. The standard error for soil isotopic composition during the experiment amounts on average 1.4‰ in natural abundance.



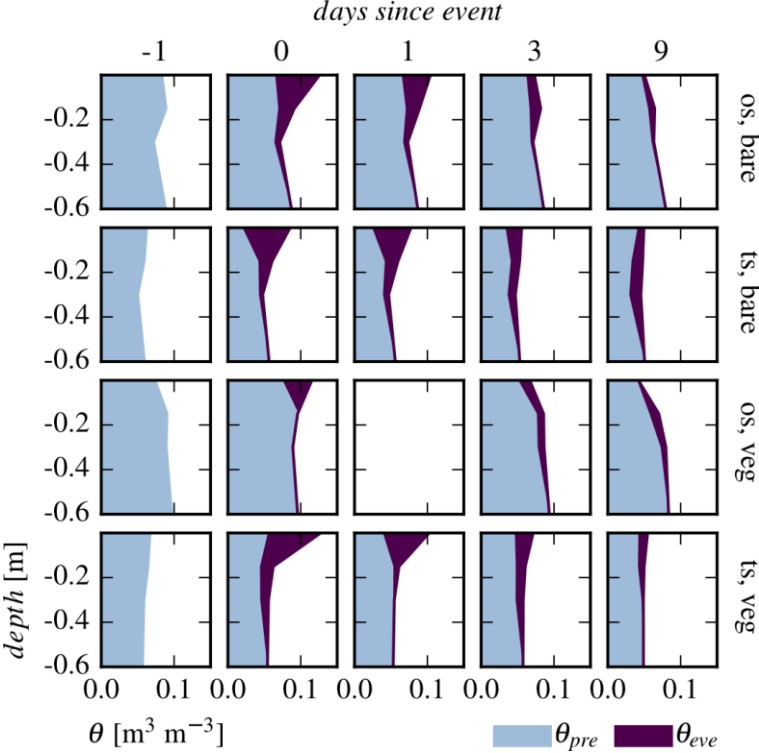

**Figure 3: Mean daily soil water content $\theta$ along soil depth separated in pre-event soil water content $\theta_{pre}$ and infiltrated event soil water content $\theta_{eve}$. Observations are displayed for plots under bare soil (bare) or vegetation cover (veg) at open sites between tree crowns (os) and shaded sites beneath tree crowns (ts). Numbers on top mark days since the watering event. Uncertainties for soil moisture observations during the experiment amount on average 2.3%$_{vol.}$ propagated from the observations. Event water partitioning for day 1 on open, vegetated plots needed to be omitted due to insufficient field data quality.**




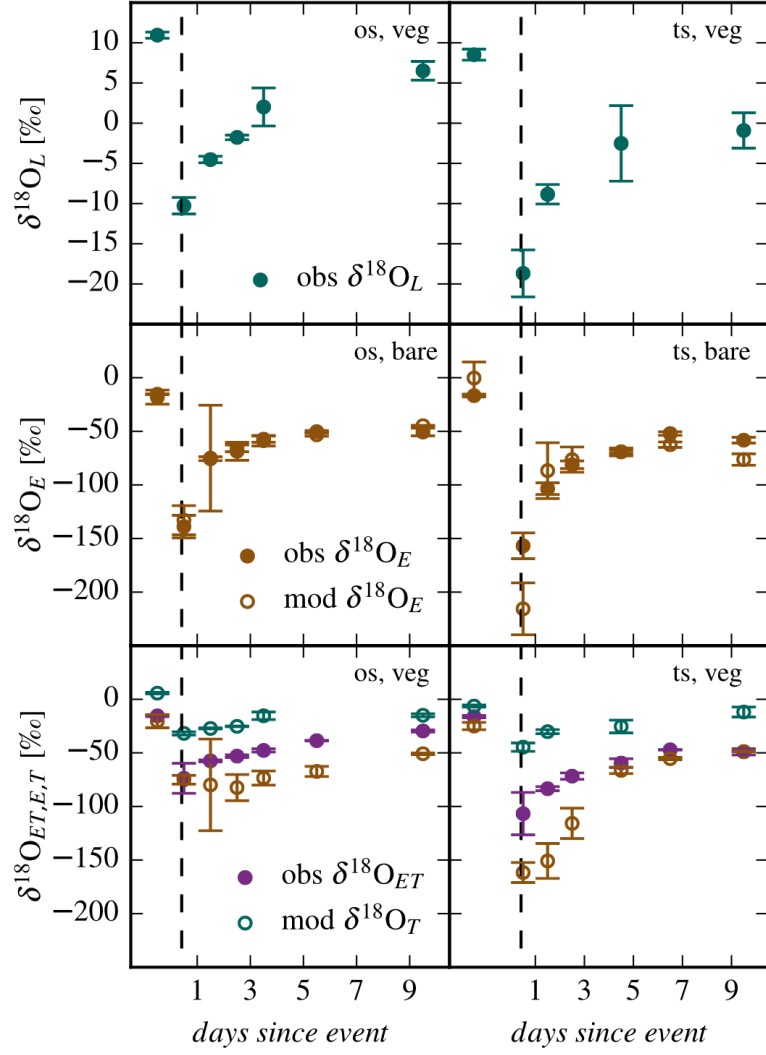

**Figure 4: Mean daily isotopic composition of bulk leaf water $\delta^{18}O_L$, soil evaporation $\delta^{18}O_E$, plant transpiration $\delta^{18}O_T$ and combined evapotranspiration $\delta^{18}O_{ET}$ from bare soil (bare) or vegetation plots (veg) at open sites between tree crowns (os) and shaded sites beneath tree crowns (ts). Full dots represent observed values (obs), hollow dots represent modelled values (mod). Dashed lines mark time of watering event. Uncertainty bars display standard error.**



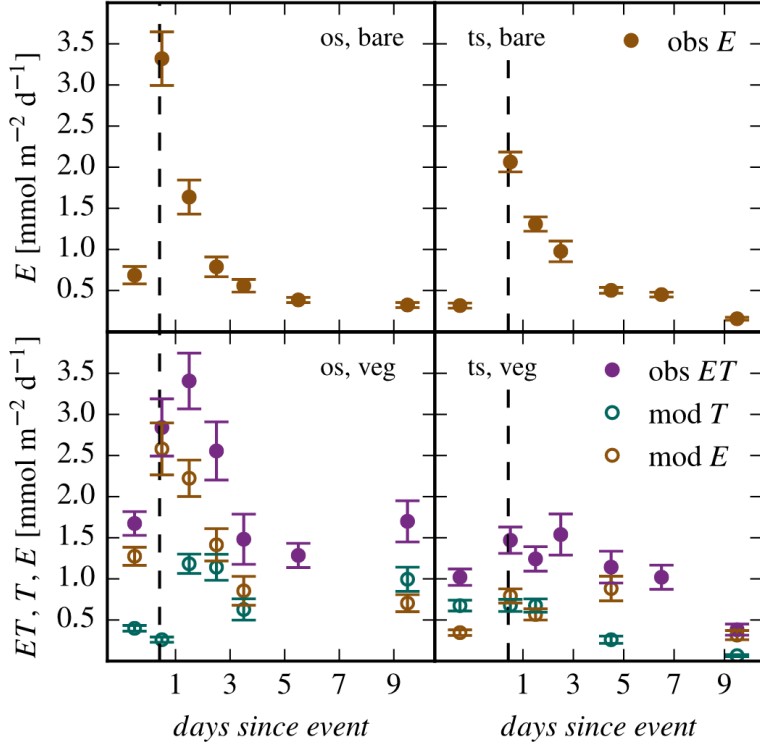

**Figure 5: Mean daily flux rates of soil evaporation *E*, plant transpiration *T* and combined evapotranspiration *ET* from bare soil (bare) or vegetation plots (veg) at open sites between tree crowns (os) and shaded sites beneath tree crowns (ts). Full dots represent observed values (obs), hollow dots represent modelled values (mod). Dashed lines mark time of watering event. Uncertainty bars display standard error.**





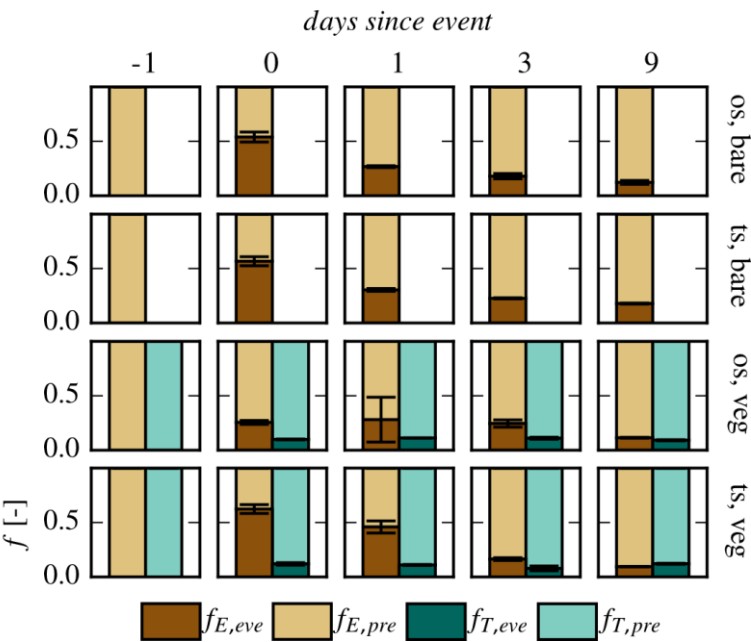

**Figure 6: Mean daily fractions $f$ of event water (*eve*) and pre-event water (*pre*) in soil evaporation $E$ and plant transpiration $T$ from bare soil (bare) or vegetation plots (veg) at open sites between tree crowns (os) and shaded sites beneath tree crowns (ts). Numbers on top mark days after watering event. Uncertainty bars display standard error.**

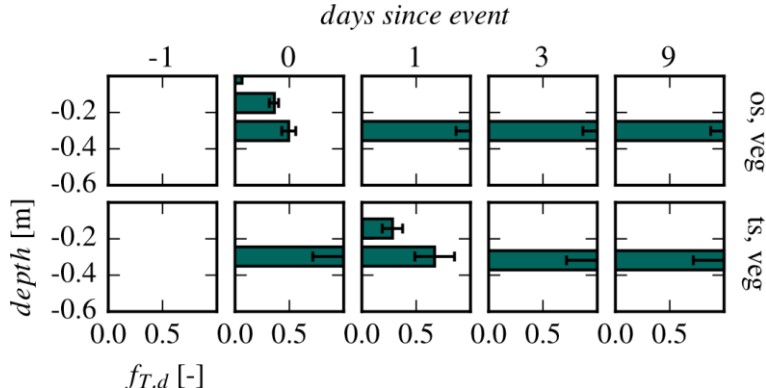

**Figure 7: Mean daily fractions of root water uptake $f_{T,d}$ of understory plants for modelled soil depths. Numbers on top mark days after watering event. Uncertainty bars display standard error.**