# Peer review of "Quantification of dynamic soil – vegetation feedbacks following an isotopically labelled precipitation pulse"

_Biogeosciences, 2016_

## Referee Comment (RC1) · Anonymous Referee #1 · 6 Dec 2016

Review Piayda and Dubbert et al., 2016 Quantification of dynamic soil – vegetation feedbacks following an isotopically labelled precipitation pulse, Biogiosciences Discuss., doi:10.5194/bg-2016-451

General Comments:

The manuscript presents a nice H218O-labelling study in a Mediterranean oak forest. Authors traced the fate of recent precipitation water in soil and understory vegetation and inferred from the respective partitions of this water for evaporation and transpiration on the use of recent precipitation for understory plants including the effects of tree shading on infiltration and water use. The study is generally well written and methods used seem generally sound. However, the discussion section at the moment is in parts

confusing and gives room for improvement, as authors discuss many theories on e.g. hydraulic lift, competition for water between trees and understory, facilitation of infiltration through tree shade etc., but presently do not relate their results very well to these theories, which at the moment hampers the conclusion that they indeed disentangled all these processes. In addition, I believe that the study would benefit from a literature evaluation on the role of tree interception on infiltration and water use, a topic that has so far been disregarded in the study. The conclusions section and the abstract at the moment include deductions that either cannot be directly seen from the results, or are not well enough discussed yet. I am confident that after revision of these issues this topical field study will be acceptable for publication and appeal to the BGS readership.

Specific Comments:

Abstract The abstract is well written, but would benefit from a revision of the conclusions.

Page 1 Line 26: "unproductive water loss" odd wording

Page 1 Line 27: this sentence should be removed, as no information on biomass production, carbon sequestration or nitrogen fixation is given in this paper

Page 1 Line 28: "Light to medium precipitation events" Only one precipitation event was studied with 20 mm. I would not consider this light or medium, also this sentence sounds as if you would compare between precipitation events of different magnitudes, which was not the case in this study.

Page 1 Line 28: "This forces plants…" Too general: In this context this sounds, as if plants were generally forced to compete for water with trees in this system. You observed only a short period of the year, for which this is probably true. Reformulate to a more differentiated conclusion considering results of this study.

Page 1 Line 33ff: a bit too thick, see comments to conclusion section

Introduction Generally nicely written, the introduction would benefit from some hypothe-
ses on tree and open side effects on water infiltration, E and T.

Page 2 Line 7: context: the use of "thus" is not indicated, I suggest removal of this term

Page 2 Line 17: context: the use of "for example" is not indicated, I suggest removal of this term

Page 2 Line 20: wording: consider rewording "stable water isotopes"

Page 2 Line 26ff: "most data sets were limited..." Some references for limited data sets would be adequate

Page 2 Line 33: "evaporative water use" Consider rewording, water that evaporates is not really used

Material and methods With small exceptions this part seems sound and methods and calculations are described adequately. However, a section on statistical analysis should be added, as the estimation of frequently mentioned significant effects in the results and discussion section cannot be inferred from the M&M part.

Page 3 Line 16: Please expand on possible effects of meshes used for bare soil plots on water infiltration

Page 3 Line 19: Irrigation was conducted how and over what time span?

Page 3 Line 28, 30 and Page 4 Line 3: replace "in a logger" by "by a logger"

Page 4 Line 6: fresh material was harvested, what was the proportion of already dry material, particularly in comparison to previous study of Dubbert et al. during a non-drought year, and the different effects of plant cover on infiltration reported in the discussion. This may have also reflected on the event water use in transpiration.

Page 4 Line 8 and 11: Presenting Fig. A1 is ok to characterize biomass and species composition differences of the sites. However, it could be redundant, as this information is only presented in the two lines here and 1 line in the discussion. Biomass and

species composition effects on event water use are not discussed much later. However, the tree site being dominated by grasses and the open site being dominated by forbs and potential effects on water use may be worth discussing, which would give presentation of this figure some more impact.

Page 4 Line 17: Calculating gt is presented as a method, but there is no data on this in the paper. I suggest removal.

Page 5 Line 5: Leaf sampling did not affect ET in the vegetation plots? How big was the reduction of leaf area through sampling? Could this have affected the temporal progress of T from event water? Please elaborate on this here.

Page 7 Line 8: depths used showed negligible root density, please add information on estimating root density in different depths to "Environmental and plant parameters"

Results This section is nicely written!

Page 8 Line 14: Consider exchanging figure numbers 3 and 4 to achieve ascending order of figures mentioned in the text.

Page 9 Line 12: correct "along with the lines of evaporation"

Page 9 Line 15: "Root water uptake allocation" sounds odd, Fig. 7 shows root water uptake from different depths over time but no allocation. Consider rephrasing.

Discussion The discussion could still be improved by further increasing the implementation of own results in the theories discussed and enhancing the clarity of some statements made.

Page 9 Line 28: remove "was"

Page 9 Line 29: add comma after "Mediterranean soils"

Page 9 Line 31: remove "significantly"

Page 10 Line 3: add "This is" bevore "in contrast"

Page 10 Line 4ff: Dubbert et al. 2014 "reported beneficial effects of vegetation cover on soil water infiltration year-round" Fig. 2 in this paper shows indeed vegetation plots showing mostly higher infiltration than soil plots. However, it would be good to compare data specifically for the transition period between the wet and the dry year here. From Fig. 2 by Dubbert et al. 2014 one can infer that vegetation enhanced infiltration as compared to bare soil, particularly with large water pulses. The only data point comparable to your data shows a rain pulse of 10mm as compared to the 20mm you gave, with only little benefit of vegetation cover. Does that enhance or reduce the significance of your reversed results? In addition, how did you apply water? On the spot irrigation can hardly be expected to yield same infiltration results as a rainfall event over a certain amount of time? This may be good to discuss here.

Page 10 Line 13: "anyway" reword

Page 10 Line 14: add "by" after "unaffected"

Page 10 Line 16: "effects of soil hydraulic properties beneath tree crowns" In what way were properties affected? Did that also apply to your study? Please elaborate further on the potential importance of this.

Page 10 Line 17: remove brackets before reference to Bhark and Small, 2003

Page 10 Line 19ff: The positive effects of tree crown cover on infiltration may be lost by interception, as the authors state. Could you try to infer the role of interception for cork oak trees from literature values to better describe the significance of the climatic advantages in the shade for infiltration? Soares David et al., 2006 for example report 22% interception loss for cork oak. (David, et al., 2006 Rainfall interception by an isolated evergreen oak tree in a Mediterranean savannah, Hydrological Processes, 20, 2713-2726; maybe also of interest: Pereira et al., 2009. Modelling interception loss from evergreen oak Mediterranean savannas: Application of a tree-based modelling approach. Agricultural and Forest Meteorology, 149(3-4): 680-688.

Page 10 Line 24: consider deleting "close to trees"

Page 10 Line 28: correct "overall"

Page 10 Line 29: reword "shortcoming", odd in this context

Page 11 Line 10: odd "productive water", consider rewording

Page 11 Line 11: rephrase to ". ... from the longer time response lag of T., on the other hand from only little event water reaching deeper soil layers, where..."

Page 11 Line 12: remove "prior to the precipitation pulse"

Page 11 Line 13: "Event water use of the understory vegetation was overall low" Again the question, of how much living biomass was there? Is it possible that understory plants were on the verge of senescence and therefore did not use the water or readjust water uptake depths?

Page 11 Line 18: "Competition with tree roots" This can this be inferred from $\delta$18O signals of soil water being more depleted in the tree site but this depletion not being visible in transpiration? Higher infiltration at the tree site must thus have been of no use for understory plants, because of competition with trees. Could you elaborate on this more?

Page 11 Line 22ff: "Hydraulic lift" This point is contrary to the previously discussed competition for water. If water from hydraulic lift was up in the layer of understory roots you would expect 1) a dilution of the event water signature, and 2) a higher soil moisture. You do not find any of this. Thus, I think from your data you can infer that hydraulic lift was not a major factor here. Roots preferentially taking up water in this depths may be due to hydraulic lift, but you find the same in the open site, so I would take out this argumentation here.

Page 12 Line 2: context: the use of "therefore" is not indicated, I suggest removal of this term

[Figure]

Page 12 Line 8: remove "the" before "type"

Conclusions The conclusions at the moment seem overstated considering the results presented, and should be rewritten. The study itself is nice enough and does not need this thick laid conclusion.

Page 12 Line 13: I do not really agree that your study disentangled and quantified tree and understory interactions. As such you compared sites with and without trees, but do not go into much depth regarding tree understory interactions. For this statement to stand this topic should be more thoroughly discussed on base of the results presented. Either adapt the discussion to really try and disentangle the role of hydraulic lift vs. competition vs. enhanced interception, or be more modest here.

Page 12 Line 18: Consider removing "or just bare soil"

Page 12 Line 19: The sentence "Thus, the amount of unproductive water loss. . . ." is a large overstatement and should be removed. This study did not show any data on nitrogen fixation, carbon sequestration or biomass production, for this statement to hold true.

Page 12 Line 21: I would not consider a 20mm precipitation pulse as light or medium.

Page 12 Line 22ff.: "Therefore, these understory plants were forced into competition. . ..However, the understory plants could profit from tree root induced soil water redistribution." Both statements do not hold true, the first point I can agree upon, but it should be included in more detail in the discussion with better implementation of own results. The second statement, I don't believe that this was shown!

Page 12 Line 23: "Cork oak trees foster infiltration. . .." I would not make this statement without considering interception of rainfall.

Page 12 Line 26: that is too laid on thick, given the study's outcome. I would not use this sentence.

Please also note the supplement to this comment:
http://www.biogeosciences-discuss.net/bg-2016-451/bg-2016-451-RC1-
supplement.pdf

---

## Referee Comment (RC2) · Anonymous Referee #2 · 13 Jan 2017

General comment

This very interesting work deals with an important and hard to assess ecohydrological problem, where once more stable isotopes prove to be useful. The manuscript presents a NICELY WELL done experiment. With very interesting results which suggests that vegetation keeps withdrawing water from the same depths after simulated rain events. Event size showed that short to medium precipitation were not very important under a dry scenario; that vegetation below trees are fierce competitors and that these lead to senescence at the beginning of the drought, and last that Trees also ameliorate the micrometeorological conditions and soil water infiltration rates. This is, in my opinion, the most relevant finding of this study. However, some issues need to be address first:

[Figure]

The authors made the experiment in a Cork-Oak forested area. However, they refer to it as Cork - Oak, cork-oak and cork oak. Please, select one and be consistent throughout the document. Please, pay attention to the use of hyphenated words.

Citations also need to be checked. For example on material and methods, the authors cite: "(Piayda et al., 2015)". However, later the authors start using parenthesis enclosing the year. I understand is possible to it like that, but for example on line 10, just before equation 4 (page 6) the citation is: "(Moreira et al., (1997); Yakir and Sternberg (2000))". However, it should read "(Moreira et al., 1997; Yakir and Sternberg, 2000)". Please, check this throughout the document. Also, pay attention to repeated parenthesis that are not needed.

Equation 2, is referenced to Craig and Gordon (1965). However, that equation does not appear in that document.

Please see attached pdf file

Where dE, stands for isotopic composition of the water vapour coming from the evaporating surface (dS) and dA stands for the atmospheric isotope composition. Also, the fractionation factor $\alpha$, is refered as $\alpha^+$, for condensation; and $\alpha^*$ for evaporation. It is important to note that in this case, and according to nomenclature introduced by Craig and Gordon, (1965), and followed by others (e.g. Gat, 1996; Gibson and Reid, 2010):

Please see attached pdf file

Please note that W and v stand for water and vapour. And that the reactant (i.e. source) is noted in last place. Hence, w-v should read as vapour to water (i.e. condensation). While, v-w should read as water to vapour (i.e. evaporation). Hence, $\alpha^*$ is used for evaporation process. I have checked also Mathieu and Bariac (1996); Dubbert et al. (2014) and couldn't find it either. Please, could you provide the right cite?; If this equation was derived by the authors, then please add include it in the appendix.

Craig H, Gordon L. 1965. Deuterium and oxygen 18 variations in the ocean and

the marine atmosphere. In Stable Isotopes in Oceanographic Studies and Paleotemperatures, Tongiorgi E (ed.).Spoleto; 9–130. which can be downloaded from http://climate.colorado.edu/research/CG/

Dubbert M, Piayda A, Cuntz M, Werner C. 2014. Oxygen isotope signatures of transpired water vapor – the role of isotopic non-steady-state transpiration of Mediterranean cork-oaks (Quercus suber L.) under natural conditions. New Phytologist 16: 2014

Gat J. 1996. Oxygen and Hydrogen isotopes in the hydrologic cycle. Annual Review of Earth and Planetary Sciences 24: 225–262. DOI: 10.1007/s13398-014-0173-7.2

Gibson J, Reid R. 2010. Stable isotope fingerprint of open-water evaporation losses and effective drainage area fluctuations in a subarctic shield watershed. Journal of Hydrology 381 (1–2): 142–150 DOI: 10.1016/j.jhydrol.2009.11.036

Mathieu R, Bariac T. 1996. A numerical model for the simulation of stable isotope profiles in drying soils. Journal of Geophysical Research 101 (D7): 12685–12696 DOI: 10.1029/96JD00223 In the reference list, please check all of them. Some of them are in full capital letters; other don't have volume and/or page number

Specific comments:

Abstract:

Check hyphenation.

Line 24 (page 1): consider using "soil evaporation and transpiration were quantified."; instead of ""evapotranspiration were quantified.". I think it would add the right value to your work, since you actually separate both evapotranspiration components.

Line 26 (page 1): it is not clear to me, who "use water"...soils or vegetation. If it refers to soils, I would change "use" for "evaporates"

Line 30 (page 1): Consider adding a comma after Thus.

Line 30 (page 1): consider rephrasing "…faster subject" to "…subjected faster"

Introduction:

Please, consider adding the hypothesis already tested in this great work. This will only add more value to your research and again, great work. Material and Methods:

Line 12 (page 3): Please, consider adding the standard deviation in the temperature and precipitation.

Line 28-29 (page 3): Please, consider rephrasing this sentence. "….was measured at 5 cm depth", instead of "…in -5 cm depth was measured".

Line 1-2 (page 4): please consider rephrasing "Volumetric soil water …."

Line 14 (page 4): Add WS to CRDS…Picarro is a Wavelenght Scanned-Cavity Ring Down Spectrometer (WS-CRDS).

Line 19 (page 4): Please, remove parenthesis enclosing the publication years, since they are not needed. Please consider separating both equations Equation 1.1 and 1.2.. For example.

Line 6 (page 5): Please, add a cite after cryogenic distillation…This will clarify which kind of system did you use…West et al., 2006 and Orlowski et al., 2013 both use cryo-distillation, but the systems are very different. Could you add also information on your water recoveries (if measured), extraction temperature and time it took the whole process of water extraction from soils and leaves. I think this will add robustness to your work.

Line 4 (Page 6): I really don't think that the mesophyll in a leaf measures 5 cm. please check the unit and correct.

Line 10 (page 6): please, remove the parenthesis from the publication years on Moreira et al. and Yakir and Sternberg.

Line 4 (page 7): please, consider "three-source linear model" instead of "three-source model".

Line 7 (page 7): please, consider removing the "s" in "depths"

Line 21 (page 7): please, consider rephrasing "(bare: 14.9 °C, veg: 11.3 °C, Fig 1)" to "(14.9° and 11.3° C for bare and vegetated soils, respectively, Fig 1)"

Line 24 (page 7): please, consider adding a comma after "Systematically. . .".

Line 8 (page 8): please, change "Lowest. . ." for "Depleted. . .", I think it is more adequate.

Line 11 (page 8): please, consider removing "only", is not needed.

Line 28 (page 8): please consider removing "here much" and adding after "than", "that of".

Please, remember that water evaporates, water is not used by evaporation or soil. (Line 10 (page 9)).

Line 23 (page 9): please check the double space you have before "Different . . ..".

Line 28 (page 9): please remove "was", not necessary.

Line 17 (page 10): please remove the parenthesis before "Bhark and Small", is not needed.

Line 6 (page 12): please add "et al" after Orlowski and remove the parenthesis from the year.

Line 8 (page 11): please remove the word "the". The word is not needed. It would be interesting that you could add more literature to this paragraph.

Craig H, Gordon L. 1965. Deuterium and oxygen 18 variations in the ocean and the marine atmosphere. In Stable Isotopes in Oceanographic Studies and Paleotemperatures, Tongiorgi E (ed.).Spoleto; 9–130.

Dubbert M, Piayda A, Cuntz M, Werner C. 2014. Oxygen isotope signatures of transpired water vapor – the role of isotopic non-steady-state transpiration of Mediterranean cork-oaks ( Quercus suber L .) under natural conditions. New Phytologist 16: 2014

Gat J. 1996. Oxygen and Hydrogen isotopes in the hydrologic cycle. Annual Review of Earth and Planetary Sciences 24: 225–262. DOI: 10.1007/s13398-014-0173-7.2

Gibson J, Reid R. 2010. Stable isotope fingerprint of open-water evaporation losses and effective drainage area fluctuations in a subarctic shield watershed. Journal of Hydrology 381 (1–2): 142–150 DOI: 10.1016/j.jhydrol.2009.11.036

Mathieu R, Bariac T. 1996. A numerical model for the simulation of stable isotope profiles in drying soils. Journal of Geophysical Research 101 (D7): 12685–12696 DOI: 10.1029/96JD00223

Line 18 (page 12): please consider changing "Irrespective" by "Regardless".

Line 22 (page 12): please consider changing "Therefore" by "Hence".

Line 23 (page 12): Do you have any proof of root water redistribution in your study area...if you have it and are planning to publish it maybe, you could briefly comment.

APENDIX A:

I would enlarge the legend, and would use the same size as the scale. Please, add a multiplication sign or a space in between g and m-2 in the Y-axis unit. As you have already made in the figures.

FIGURES:

On figures 2, 4 and 5: too much iteration in the X-axis "days since event". I would leave only one and change it for (time (d)) or simply "days"; and in the caption I would mention before that the dashed line is the watering event (on figure 2)

Why are all axis in italics?, would be better to have them normal?...just a thought.

On figures 4 and 5: I would try to use a different symbol...squares, triangles, and diamonds. I think this will improve the readability of the figures. Please, this last is only a suggestion. However, I think it will improve the readability and impact of, again, such a nice work.

Figures 6 and 7: please, center the Y-axis name.

Please also note the supplement to this comment:
http://www.biogeosciences-discuss.net/bg-2016-451/bg-2016-451-RC2-supplement.pdf

---

## Author Response (AR1)

**Final author response to Anonymous Referee # 1 (RC1)**

*Referee comments* – Author response

**General Comments:**

*The manuscript presents a nice $H_2^{18}O$-labelling study in a Mediterranean oak forest. Authors traced the fate of recent precipitation water in soil and understory vegetation and inferred from the respective partitions of this water for evaporation and transpiration on the use of recent precipitation for understory plants including the effects of tree shading on infiltration and water use. The study is generally well written and methods used seem generally sound. However, the discussion section at the moment is in parts confusing and gives room for improvement, as authors discuss many theories on e.g. hydraulic lift, competition for water between trees and understory, facilitation of infiltration through tree shade etc., but presently do not relate their results very well to these theories, which at the moment hampers the conclusion that they indeed disentangled all these processes. In addition, I believe that the study would benefit from a literature evaluation on the role of tree interception on infiltration and water use, a topic that has so far been disregarded in the study. The conclusions section and the abstract at the moment include deductions that either cannot be directly seen from the results, or are not well enough discussed yet. I am confident that after revision of these issues this topical field study will be acceptable for publication and appeal to the BGS readership.*

The authors are thankful for the general appreciation of the submitted manuscript and the recommendation for publication in *Biogeosciences* by Anonymous referee # 1. The authors highly appreciate the thorough review of the manuscript and the very constructive comments. The authors have reviewed the manuscript with special focus on the discussion and conclusion section and include the mentioned literature evaluation on the role of interception.

**Specific Comments:**

**Abstract**

*The abstract is well written, but would benefit from a revision of the conclusions.*

The authors are thankful for the appreciation of the referee and incorporated the revised conclusions in the abstract.

*Page 1 Line 26: "unproductive water loss" odd wording*

Changed to unproductive evaporation.

*Page 1 Line 27: this sentence should be removed, as no information on biomass production, carbon sequestration or nitrogen fixation is given in this paper*

The sentence was removed.

*Page 1 Line 28: "Light to medium precipitation events" Only one precipitation event was studied with 20 mm. I would not consider this light or medium, also this sentence sounds as if you would compare between precipitation events of different magnitudes, which was not the case in this study.*

The authors agree that 20 mm of rain during one hour of watering can be considered as high precipitation intensity compared to the natural precipitation regime of the study site. We can

consequentially be very certain about the fact, that naturally occurring light to medium precipitation events during drought periods have no effect on root water uptake, since the high precipitation intensity of the experiment had little impact either. We omitted the latter part of the sentence.

*Page 1 Line 28: "This forces plants..." Too general: In this context this sounds, as if plants were generally forced to compete for water with trees in this system. You observed only a short period of the year, for which this is probably true. Reformulate to a more differentiated conclusion considering results of this study.*

The statement was related to the drought period of the experiment and the onset of summer.

*Page 1 Line 33ff: a bit too thick, see comments to conclusion section*

The sentence was shortened.

**Introduction**

*Generally nicely written, the introduction would benefit from some hypotheses on tree and open side effects on water infiltration, E and T.*

The authors are thankful for the appreciation of the referee. The authors agree that working hypotheses will enhance the structure of the manuscript and incorporated the following hypotheses in the introduction, discussion and conclusions:

I. Presence of understory vegetation increases evapotranspirative water loss compared to bare soil, but foster infiltration due to shading.
II. Preferential root water uptake depth of understory plants is unaffected by changes in soil water availability after rain pulses during drought.
III. Tree shading fosters infiltration of event water and reduces evapotranspiration generating favourable soil moisture conditions for understory plants.

*Page 2 Line 7: context: the use of "thus" is not indicated, I suggest removal of this term*

The term was removed.

*Page 2 Line 17: context: the use of "for example" is not indicated, I suggest removal of this term*

The term was removed.

*Page 2 Line 20: wording: consider rewording "stable water isotopes"*

The authors consider "stable water isotopes" as a common term for $D_2O^{16}$ and $H_2O^{18}$ isotopes in literature (c.f. *Sturm et al.* An introduction to stable water isotopes in climate models: benefits of forward proxy modelling for paleoclimatology, Climate of the Past, 2010) and insist of using it consistently with existing scientific publications.

*Page 2 Line 26ff: "most data sets were limited..." Some references for limited data sets would be adequate*

We now cite the works of Kurz-Besson et al., 2006 and Asbjörnsen et al., 2008

*Page 2 Line 33: "evaporative water use" Consider rewording, water that evaporates is not really used*

The term was changed to soil evaporation.

**Material and methods**

*With small exceptions this part seems sound and methods and calculations are described adequately. However, a section on statistical analysis should be added, as the estimation of frequently mentioned significant effects in the results and discussion section cannot be inferred from the M&M part.*

The authors are thankful for the appreciation of the referee. Section 2.8 was added, reporting the error propagation to the results as follows: All results are reported as replicate mean with associated standard error to achieve comparability between different sample sizes. All model calculations were applied to single replica and averaged afterwards. Observed effects were considered statistically different when no overlap of standard errors was observed.

*Page 3 Line 16: Please expand on possible effects of meshes used for bare soil plots on water infiltration*

The requested information was added: meshes were installed vertically, circumventing the undisturbed soil. The sites were kept vegetation free just by regular weeding. We expect no influence of the mesh on infiltration, since the plots were installed one year before the experiment and processes like preferential flow along the mesh is unlikely.

*Page 3 Line 19: Irrigation was conducted how and over what time span?*

The requested information was added: After a base line observation, all plots were watered with 20 mm water within one hour using watering cans. The water showed an oxygen isotopic signature of -139.5‰ to trace the influence of different vegetation components on water infiltration. All plots and the surrounding soil were watered equally to avoid lateral gradients and possible differences between trenched and control plots.

*Page 3 Line 28, 30 and Page 4 Line 3: replace "in a logger" by "by a logger"*

The term was corrected.

*Page 4 Line 6: fresh material was harvested, what was the proportion of already dry material, particularly in comparison to previous study of Dubbert et al. during a non-drought year, and the different effects of plant cover on infiltration reported in the discussion. This may have also reflected on the event water use in transpiration.*

In this particular year the proportion of dry material was minimal owing to the fact that due to the additional severe dry period between January and March 2012 the biomass development in general was very low and developed only following the start of the drought release in March. Dead biomass from the previous season was removed from the plots at the end of summer 2011.

*Page 4 Line 8 and 11: Presenting Fig. A1 is ok to characterize biomass and species composition differences of the sites. However, it could be redundant, as this information is only presented in the two lines here and 1 line in the discussion. Biomass and species composition effects on event water use are not discussed much later. However, the tree site being dominated by grasses and the open site being dominated by forbs and potential effects on water use may be worth discussing, which would give presentation of this figure some more impact.*

We agree and now discuss this effect in the discussion section (see page 12 line 32 to page 13 line 9)

*Page 4 Line 17: Calculating gt is presented as a method, but there is no data on this in the paper. I suggest removal.*

The sentence was shortened by removing total conductance.

*Page 5 Line 5: Leaf sampling did not affect ET in the vegetation plots? How big was the reduction of leaf area through sampling? Could this have affected the temporal progress of T from event water? Please elaborate on this here.*

This is a very important issue indeed. Our leaf sampling protocol did ensure that leaf biomass sampling for isotope analysis was affecting the overall living biomass to an extend less than 5%, as we did not sample species specifically but took representative samples of the vegetation. Accordingly, we argue that the effects of destructive sampling were minimal in particular regarding the effect of event water use.

*Page 7 Line 8: depths used showed negligible root density, please add information on estimating root density in different depths to "Environmental and plant parameters"*

Below ground biomass was sampled with soil cores in -5, -15, -30, and -60 cm depth. Oven dried soil was sieved and root biomass was determined gravimetrically. 80 % of root biomass was distributed between -5 to -15 cm depth. Only 5% was distributed above -5 cm and 15% between -20 to -35 cm depth.

**Results**

*This section is nicely written!*

The authors are thankful for the appreciation of the referee.

*Page 8 Line 14: Consider exchanging figure numbers 3 and 4 to achieve ascending order of figures mentioned in the text.*

The authors ordered the figures 2 and 3 (we assume that the referee was not referring to 3 and 4, since they are not mentioned in the particular position of the manuscript) from measured to modelled data in order to show results in a logical order of retrieval. We therefore keep the current ordering.

*Page 9 Line 12: correct "along with the lines of evaporation"*

The term was corrected.

*Page 9 Line 15: "Root water uptake allocation" sounds odd, Fig. 7 shows root water uptake from different depths over time but no allocation. Consider rephrasing.*

The term was rephrased in the entire manuscript to "preferential root water uptake depth".

**Discussion**

*The discussion could still be improved by further increasing the implementation of own results in the theories discussed and enhancing the clarity of some statements made.*

We appreciate the constructive suggestions and revised the discussion section in accordance with the suggestions.

*Page 9 Line 28: remove "was"*

The term was removed.

*Page 9 Line 29: add comma after "Mediterranean soils"*

The sentence was corrected.

*Page 9 Line 31: remove "significantly"*

The sentence was corrected.

*Page 10 Line 3: add "This is" bevore "in contrast"*

The sentence was corrected.

*Page 10 Line 4ff: Dubbert et al. 2014 "reported beneficial effects of vegetation cover on soil water infiltration year-round" Fig. 2 in this paper shows indeed vegetation plots showing mostly higher infiltration than soil plots. However, it would be good to compare data specifically for the transition period between the wet and the dry year here. From Fig. 2 by Dubbert et al. 2014 one can infer that vegetation enhanced infiltration as compared to bare soil, particularly with large water pulses. The only data point comparable to your data shows a rain pulse of 10mm as compared to the 20mm you gave, with only little benefit of vegetation cover. Does that enhance or reduce the significance of your reversed results? In addition, how did you apply water? On the spot irrigation can hardly be expected to yield same infiltration results as a rainfall event over a certain amount of time? This may be good to discuss here.*

The precipitation data displayed in Dubbert et al. 2014 (Fig. 2) represents daily sums of precipitation. Even though the daily sum of precipitation at the comparable data point end of May shows 10 mm of rain, the precipitation intensity could have been very different to the experiment conducted in this study. A low intensity of, e.g. 1mm per hour, would change soil moisture conditions and air moisture conditions in the boundary layer in the very beginning, fostering different processes during infiltration for the last 9 hours of the event. The results shown here are only valid for short term rain events with high intensities and thus not contradictory to the results of Dubbert et al. 2014. However, the authors agree with the referee that the topic of intensities need to be discussed. The authors changed the respective discussion section to: This is in contrast to previous studies, which reported beneficial effects of plant cover on daily sum of infiltration during the same period at the onset of drought in 2011 (Dubbert et al., 2014c). However, (Dubbert et al., 2014c) only observed precipitation events of light intensity during the period of interest. The present study reports on high intensity precipitation events. This unexpected turn in effect direction with increasing precipitation intensity, which depends on plant cover and atmospheric evapotranspirative demand, potentially plays a strong role for the water balance of the ecosystem in the course of ongoing climate change scenarios since the occurrence of extreme precipitation events is expected to increase (IPCC, 2013).

*Page 10 Line 13: "anyway" reword*

The sentence was corrected.

*Page 10 Line 14: add "by" after "unaffected"*

The sentence was corrected.

*Page 10 Line 16: "effects of soil hydraulic properties beneath tree crowns" In what way were properties affected? Did that also apply to your study? Please elaborate further on the potential importance of this.*

The respective discussion section was changed to: Previous studies reported similar, positive feedbacks of tree cover for the hydrological cycle in savannah-type ecosystems related to shading effects (Eldridge and Freudenberger, 2005). Effects of altered soil hydraulic properties beneath tree crowns, like the amount of preferential flow fostering infiltration (Bargués Tobella et al., 2014) could not be identified in this study.

*Page 10 Line 17: remove brackets before reference to Bhark and Small, 2003*

The sentence was corrected.

*Page 10 Line 19ff: The positive effects of tree crown cover on infiltration may be lost by interception, as the authors state. Could you try to infer the role of interception for cork oak trees from literature values to better describe the significance of the climatic advantages in the shade for infiltration?¡*

The amount of interception loss by the tree canopy and stem bark of cork-oaks (or trees in general) is highly variable, depending on meteorological variables like precipitation intensity, wind speed, relative air moisture and stand properties like tree density, branch geometry, leaf angle and shape. The authors included results from David et al. 2006 in the discussion, which were derived in an ecosystem with comparable stand and climatic conditions in order to give the reader a feeling for the magnitudes of the interception loss and infiltration enhancement. However, directly relating tree interception loss results from other studies to the infiltration effect results of this study is highly prone to misleading conclusions due to different boundary conditions and settings of the experiments. The authors therefore desist from direct deductions by comparisons with previous studies.

*Page 10 Line 24: consider deleting "close to trees"*

The term was deleted.

*Page 10 Line 28: correct "overall"*

The sentence was corrected.

*Page 10 Line 29: reword "shortcoming", odd in this context*

The term was replaced by negative effect.

*Page 11 Line 10: odd "productive water", consider rewording*

The term was replaced by transpiration water.

*Page 11 Line 11: rephrase to ".... from the longer time response lag of T., on the other hand from only little event water reaching deeper soil layers, where..."*

The sentence was reformulated.

*Page 11 Line 12: remove "prior to the precipitation pulse"*

The term was removed.

*Page 11 Line 13: "Event water use of the understory vegetation was overall low" Again the question, of how much living biomass was there? Is it possible that understory plants were on the verge of senescence and therefore did not use the water or readjust water uptake depths?*

At both sites the understory vegetation was indeed already past the peak of biomass development. There were, however, differences between the two regarding the productivity evolving during the experimental period. At the open site, the understory still showed a significant net uptake of carbon throughout the entire experiment, while decreasing NEE and even a net release of carbon at the final day of the experiment could be observed at the tree site. Since we agree, that this information is rather important for the interpretation of the site specific difference and also explains the overall differences in ET and T throughout the experiment rather well, we added an additional graph A2, informing on the development of NEE over the experimental course. See also page 12 line 32 to page 13 line 9.

*11      18:                                                                     18O signals of soil water being more depleted in the tree site but this depletion not being visible in transpiration? Higher infiltration at the tree site must thus have been of no use for understory plants, because of competition with trees. Could you elaborate on this more?*

It is indeed true that leaf and transpirative isotopic signatures did not show a such significant depletion at the tree site compared to the open site as could be observed for the soil isotopic values. This is mostly due to the lesser general uptake of water (i.e. lower T rate) below the trees compared to the open site. Whether this is due to competition with trees is not provable with the current data set, mostly because we are missing isotopic data on tree root water uptake (tree xylem). Moreover, the current approach of spacially explicit labelling of the discreet plots did not allow for estimation of tree reaction to the irrigation pulse.

What can be clearly seen is, that the vegetation below the trees was already at the verge of senescence (see above). Previous data by Dubbert et al. (2014) however suggests, that the phenological shift and earlier senescence might very well be strongly related to tree understory competition.

*Page 11 Line 22ff: "Hydraulic lift" This point is contrary to the previously discussed competition for water. If water from hydraulic lift was up in the layer of understory roots you would expect 1) a dilution of the event water signature, and 2) a higher soil moisture. You do not find any of this. Thus, I think from your data you can infer that hydraulic lift was not a major factor here. Roots preferentially taking up water in this depths may be due to hydraulic lift, but you find the same in the open site, so I would take out this argumentation here.*

The authors agree with the opinion of the referee and removed this discussion section.

*Page 12 Line 2: context: the use of "therefore" is not indicated, I suggest removal of this term*

The term was removed.

*Page 12 Line 8: remove "the" before "type"*

The sentence was corrected.

**Conclusions**

*The conclusions at the moment seem overstated considering the results presented, and should be rewritten. The study itself is nice enough and does not need this thick laid conclusion.*

The authors incorporated the recommendations of the referee into the conclusions section to a large degree in order to make it more adequate for results presented in the study.

*Page 12 Line 13: I do not really agree that your study disentangled and quantified tree and understory interactions. As such you compared sites with and without trees, but do not go into much depth regarding tree understory interactions. For this statement to stand this topic should be more thoroughly discussed on base of the results presented. Either adapt the discussion to really try and disentangle the role of hydraulic lift vs. competition vs. enhanced interception, or be more modest here.*

The authors reformulated the sentence: In this study, the various interactions between understory vegetation and trees of a Mediterranean cork – oak woodland affecting the ecosystem water flows could be quantified.

*Page 12 Line 18: Consider removing "or just bare soil"*

The term was removed.

*Page 12 Line 19: The sentence "Thus, the amount of unproductive water loss...." is a large overstatement and should be removed. This study did not show any data on nitrogen fixation, carbon sequestration or biomass production, for this statement to hold true.*

The sentence was modified to: Thus, the amount of unproductive evaporation is largely reduced, in favour of transpiration.

*Page 12 Line 21: I would not consider a 20mm precipitation pulse as light or medium.*

The sentence was changed accordingly.

*Page 12 Line 22ff.: "Therefore, these understory plants were forced into competition....However, the understory plants could profit from tree root induced soil water redistribution." Both statements do not hold true, the first point I can agree upon, but it should be included in more detail in the discussion with better implementation of own results. The second statement, I don't believe that this was shown!*

The statement was removed.

*Page 12 Line 23: "Cork oak trees foster infiltration...." I would not make this statement without considering interception of rainfall.*

The sentence states that the study could show a strong increase of infiltration due to favourable climatic conditions under tree crowns. That is true independent of a possible negative effect of interception losses on throughfall. However, the authors agree with the referee that the effect of

throughfall interception was not investigated in this study and plays a major role in the overall ecosystem water balance. Therefore the authors keep this important statement, but reworded the sentence by removing the emphasis of the infiltration part in the revised version.

*Page 12 Line 26: that is too laid on thick, given the study's outcome. I would not use this sentence.*

The sentence was removed.

**Final author response to Anonymous Referee # 2 (RC2)**

*Referee comments* – Author response

**General Comments:**

*This very interesting work deals with an important and hard to assess ecohydrological problem, where once more stable isotopes prove to be useful. The manuscript presents a NICELY WELL done experiment. With very interesting results which suggests that vegetation keeps withdrawing water from the same depths after simulated rain events. Event size showed that short to medium precipitation were not very important under a dry scenario; that vegetation below trees are fierce competitors and that these lead to senescence at the beginning of the drought, and last that Trees also ameliorate the micrometeorological conditions and soil water infiltration rates. This is, in my opinion, the most relevant finding of this study.*

*However, some issues need to be address first: The authors made the experiment in a Cork-Oak forested area. However, they refer to it as Cork - Oak, cork-oak and cork oak. Please, select one and be consistent throughout the document. Please, pay attention to the use of hyphenated words.*

The manuscript have been revised for consistent naming and use of hyphenation.

*Citations also need to be checked. For example on material and methods, the authors cite: "(Piayda et al., 2015)". However, later the authors start using parenthesis enclosing the year. I understand is possible to it like that, but for example on line 10, just before equation 4 (page 6) the citation is: "(Moreira et al., (1997); Yakir and Sternberg (2000))". However, it should read "(Moreira et al., 1997; Yakir and Sternberg, 2000)". Please, check this throughout the document. Also, pay attention to repeated parenthesis that are not needed.*

We checked the citations list and citations within the text and corrected the errors. We apologize for the inconvenience.

*Equation 2, is referenced to Craig and Gordon (1965). However, that equation does not appear in that document.*

$$\delta_E = \frac{1}{(1-h)+\Delta\varepsilon}\left(\frac{\delta_S}{\alpha_{v-w}^*} - (h\delta_A) - \Delta\varepsilon + \varepsilon^*\right)$$

*Where dE, stands for isotopic composition of the water vapour coming from the evaporating surface (dS) and dA stands for the atmospheric isotope composition. Also, the fractionation factor α, is refered as α + , for condensation; and α ∗ for evaporation. It is important to note that in this case, and according to nomenclature introduced by Craig and Gordon, (1965), and followed by others (e.g. Gat, 1996; Gibson and Reid, 2010):*

$$\alpha_{w-v}^+ = \frac{1}{\alpha_{v-w}^+}$$

$$\frac{1}{\alpha_{v-w}^+} = \alpha_{v-w}^*$$

*Please note that W and v stand for water and vapour. And that the reactant (i.e. source) is noted in last place. Hence, w-v should read as vapour to water (i.e. condensation). While, v-w should read as*

*water to vapour (i.e. evaporation). Hence, α ∗ is used for evaporation process. I have checked also Mathieu and Bariac (1996); Dubbert et al. (2014) and couldn't find it either. Please, could you provide the right cite?; If this equation was derived by the authors, then please add include it in the appendix.*

*Craig H, Gordon L. 1965. Deuterium and oxygen 18 variations in the ocean and the marine atmosphere. In Stable Isotopes in Oceanographic Studies and Paleotemperatures, Tongiorgi E (ed.).Spoleto; 9–130. which can be downloaded from http://climate.colorado.edu/research/CG/*

*Dubbert M, Piayda A, Cuntz M, Werner C. 2014. Oxygen isotope signatures of transpired water vapor – the role of isotopic non-steady-state transpiration of Mediterranean cork-oaks (Quercus suber L.) under natural conditions. New Phytologist 16: 2014*

*Gat J. 1996. Oxygen and Hydrogen isotopes in the hydrologic cycle. Annual Review of Earth and Planetary Sciences 24: 225–262. DOI: 10.1007/s13398-014-0173-7.2*

*Gibson J, Reid R. 2010. Stable isotope fingerprint of open-water evaporation losses and effective drainage area fluctuations in a subarctic shield watershed. Journal of Hydrology 381 (1–2): 142–150 DOI:10.1016/j.jhydrol.2009.11.036*

*Mathieu R, Bariac T. 1996. A numerical model for the simulation of stable isotope profiles in drying soils. Journal of Geophysical Research 101 (D7): 12685–12696 DOI: 10.1029/96JD00223*

*In the reference list, please check all of them. Some of them are in full capital letters; other don't have volume and/or page number*

We apologize for the errors in the citation list, they are checked and corrected. The Craig and Gordon formula was written however not in delta notation but isotope ratios following previous publication of the authors. We now cite Dubbert et al., 2013 and Harverd and Cuntz, 2010 to refer to it. In addition we added a sentence on the transformation of $R_E$ to $\delta_E$ (page 6 line 7).

**Specific Comments:**

**Abstract**

*Check hyphenation.*

The hyphenation errors have been corrected.

*Line 24 (page 1): consider using "soil evaporation and transpiration were quantified."; instead of ""evapotranspiration were quantified.". I think it would add the right value to your work, since you actually separate both evapotranspiration components.*

The sentence was changed accordingly.

*Line 26 (page 1): it is not clear to me, who "use water"…soils or vegetation. If it refers to soils, I would change "use" for "evaporates"*

The term refers to transpiration by plants and was changed accordingly.

*Line 30 (page 1): Consider adding a comma after Thus.*

The sentence was corrected.

*Line 30 (page 1): consider rephrasing "...faster subject" to "...subjected faster"*

The sentence was changed accordingly.

**Introduction**

*Please, consider adding the hypothesis already tested in this great work. This will only add more value to your research and again, great work.*

The authors are thankful for the appreciation of the referee. The authors agree that working hypotheses will enhance the structure of the manuscript and incorporated the following hypotheses in the introduction, discussion and conclusions:

I. Presence of understory vegetation increases evaptranspirative water loss compared to bare soil, but foster infiltration due to shading.
II. Preferential root water uptake depth of understory plants is unaffected by changes in soil water availability after rain pulses during drought.
III. Tree shading fosters infiltration of event water and reduces evapotranspiration generating favourable soil moisture conditions for understory plants.

**Material and Methods**

*Line 12 (page 3): Please, consider adding the standard deviation in the temperature and precipitation.*

The authors do not have access to data about the standard deviation of the long term temperature and precipitation distribution and therefore apologize for the missing information.

*Line 28-29 (page 3): Please, consider rephrasing this sentence. "....was measured at 5 cm depth", instead of "...in -5 cm depth was measured".*

The sentence was changed accordingly.

*Line 1-2 (page 4): please consider rephrasing "Volumetric soil water ...."*

The sentence was changed accordingly.

*Line 14 (page 4): Add WS to CRDS...Picarro is a Wavelenght Scanned-Cavity Ring Down Spectrometer (WS-CRDS).*

The sentence was changed accordingly.

*Line 19 (page 4): Please, remove parenthesis enclosing the publication years, since they are not needed. Please consider separating both equations Equation 1.1 and 1.2.. For example.*

The parenthesis were removed and the equation was split in two.

*Line 6 (page 5): Please, add a cite after cryogenic distillation...This will clarify which kind of system did you use...West et al., 2006 and Orlowski et al., 2013 both use cryo-distillation, but the systems are very different. Could you add also information on your water recoveries (if measured), extraction temperature and time it took the whole process of water extraction from soils and leaves. I think this will add robustness to your work.*

We used a cryogenic system of our own design, which are in long term use in the labs in the PSI and Freiburg. The system is similar to that of Orlowski et al., which we cite now respectively.

*Line 4 (Page 6): I really don't think that the mesophyll in a leaf measures 5 cm. please check the unit and correct.*

This was misleading. The 0.05 m refers to the effective path length. We corrected the sentences.

*Line 10 (page 6): please, remove the parenthesis from the publication years on Moreira et al. and Yakir and Sternberg.*

The sentence was corrected.

*Line 4 (page 7): please, consider "three-source linear model" instead of "three-source model".*

The term was changed accordingly.

*Line 7 (page 7): please, consider removing the "s" in "depths"*

The term was corrected.

*Line 21 (page 7): please, consider rephrasing "(bare: 14.9 °C, veg: 11.3 °C, Fig 1)" to "(14.9° and 11.3° C for bare and vegetated soils, respectively, Fig 1)"*

The term was changed accordingly.

**Results**

*Line 24 (page 7): please, consider adding a comma after "Systematically...".*

The sentence was corrected.

*Line 8 (page 8): please, change "Lowest..." for "Depleted...", I think it is more adequate.*

The term was changed accordingly.

*Line 11 (page 8): please, consider removing "only", is not needed.*

The term was removed.

*Line 28 (page 8): please consider removing "here much" and adding after "than", "that of". Please, remember that water evaporates, water is not used by evaporation or soil. (Line 10 (page 9)).*

The sentence was changed accordingly.

**Discussion**

*Line 23 (page 9): please check the double space you have before "Different ....".*

The space was removed.

*Line 28 (page 9): please remove "was", not necessary.*

The term was removed.

*Line 17 (page 10): please remove the parenthesis before "Bhark and Small", is not needed.*

The parenthesis was removed.

*Line 6 (page 12): please add "et al" after Orlowski and remove the parenthesis from the year.*

The citation was corrected.

*Line 8 (page 11): please remove the word "the". The word is not needed. It would be interesting that you could add more literature to this paragraph.*

*Craig H, Gordon L. 1965. Deuterium and oxygen 18 variations in the ocean and the marine atmosphere. In Stable Isotopes in Oceanographic Studies and Paleotemperatures, Tongiorgi E (ed.).Spoleto; 9–130.*

*Dubbert M, Piayda A, Cuntz M, Werner C. 2014. Oxygen isotope signatures of transpired water vapor – the role of isotopic non-steady-state transpiration of Mediterranean cork-oaks ( Quercus suber L .) under natural conditions. New Phytologist 16: 2014*

*Gat J. 1996. Oxygen and Hydrogen isotopes in the hydrologic cycle. Annual Review of Earth and Planetary Sciences 24: 225–262. DOI: 10.1007/s13398-014-0173-7.2*

*Gibson J, Reid R. 2010. Stable isotope fingerprint of open-water evaporation losses and effective drainage area fluctuations in a subarctic shield watershed. Journal of Hydrology 381 (1–2): 142–150 DOI: 10.1016/j.jhydrol.2009.11.036*

*Mathieu R, Bariac T. 1996. A numerical model for the simulation of stable isotope profiles in drying soils. Journal of Geophysical Research 101 (D7): 12685–12696 DOI: 10.1029/96JD00223*

The term was removed. Regarding the literature no specific action was taken. However, in response to the suggestions of both reviewers, the paragraph was restructured and more literature was added in response to the other specific comments.

**Conclusion**

*Line 18 (page 12): please consider changing "Irrespective" by "Regardless".*

The term was changed accordingly.

*Line 22 (page 12): please consider changing "Therefore" by "Hence".*

The term was changed accordingly.

*Line 23 (page 12): Do you have any proof of root water redistribution in your study area…if you have it and are planning to publish it maybe, you could briefly comment.*

Unfortunately we do not have data on root water redistribution at our study sites. Hence, the authors removed the aspect of root water redistribution from the discussion in compliance with important comments of referee RC1.

**Quantification of dynamic soil-vegetation feedbacks following an isotopically labelled precipitation pulse**

Arndt Piayda*,[1], Maren Dubbert*,[2], Rolf Siegwolf[3], Matthias Cuntz[4], Christiane Werner[2]

*equal contribution

[1]Thünen Institute of Climate-Smart Agriculture, Braunschweig, 38116, Germany
[2]Ecosystem Physiology, University Freiburg, Freiburg, 79110, Germany
[3]Lab for Atmospheric Chemistry, Ecosystems and Stable Isotope Research, Paul Scherrer Institut, Villingen PSI, 5232, Switzerland
[4]UMR Ecologie et Ecophysiologie Forestières, UMR1137, INRA-Université de Lorrain, Champenoux-54500 Vandoeuvre Les Nancy, 54280, France

*Correspondence to*: Arndt Piayda (arndt.piayda@thuenen.de), Maren Dubbert (maren.dubbert@cep.uni-freiburg.de)

**Abstract.** The presence of vegetation alters hydrological cycles of ecosystems. Complex plant-soil interactions govern the fate of precipitation input and water transitions through ecosystem compartments. Disentangling these interactions is a major challenge in the field of ecohydrology and pivotal foundation for understanding the carbon cycle of semi-arid ecosystems. Stable water isotopes can be used in this context as tracer to quantify water movement through soil-vegetation-atmosphere interfaces.

The aim of this study is to disentangle vegetation effects on soil water infiltration and distribution as well as dynamics of soil evaporation and grassland water use in a Mediterranean cork oak woodland during dry conditions. An irrigation experiment using $\delta^{18}O$ labeled water was carried out in order to quantify distinct effects of tree and herbaceous vegetation on infiltration and distribution of event water in the soil profile. Dynamic responses of soil and herbaceous vegetation fluxes to precipitation regarding event water use, water uptake depth plasticity and contribution to ecosystem soil evaporation and transpiration were quantified.

Total water loss to the atmosphere from bare soil was as high as from vegetated soil, utilizing large amounts of unproductive evaporation for transpiration, but infiltration rates decreased. No adjustments of main root water uptake depth to changes of water availability could be observed during the experiment. This forces understory plants to compete with adjacent trees for  water in deeper soil layers at the onset of summer. Thus, understory plants are subjected faster  to chronic water deficits, leading to premature senescence at the onset of drought. Despite this water competition, the presence of cork oak trees fosters infiltration  and reduces  evapotranspirative water losses from understory and soil, both due to altered micro climatic conditions under  crown shading. This study highlights

complex soil—plant—atmosphere and inter—species interactions  controlling  rain pulse transitions through a typical Mediterranean savannah ecosystem, disentangled by the use of stable water isotopes.

**1 Introduction**

[revised manuscript text omitted]

I. Presence of understory vegetation increases evapotranspirative water loss compared to bare soil, but foster infiltration due to shading.

II. Preferential root water uptake depth of understory plants is unaffected by changes in soil water availability after rain pulses during drought.

III. Tree shading fosters infiltration of event water and reduces evapotranspiration generating favourable soil moisture conditions for understory plants.

**2 Material and methods**

**2.1 Study site and experimental design**

Measurements were conducted between May 26 and June 6 2012 in an open cork-_oak woodland (*Quercus suber* L.) in central Portugal, approximately 100 km north-east of Lisbon (N39°8'17.84'' W8°20'3.76''; Herdade de Machoqueira do Grou). The trees are widely spaced (209 individuals ha$^{-1}$) with a leaf area index of 1.1 and a gap probability of 0.7 (Piayda et al., 2015).

The herbaceous layer is dominated by native annual forbs and grasses. The site is characterized by Mediterranean climate, with a 30 year long-term mean annual temperature of approximately 15.9 °C and annual precipitation of 680 mm (Instituto de Meteorologia, Lisbon). We established two sites: one directly under the oak crown projected area (tree site, ts) and another one in an adjacent open area (open site, os). Two types of plots (sized 40 × 80 cm) were installed in each site: bare soil plots with total exclusion of above and below--ground biomass (lateral root ingrowth was prevented by vertically inserted trenching meshes around the plots, mesh diameter < 1 μm, Plastok, Birkenhead, UK), and understory plots with herbaceous vegetation (four plots per site and treatment). All plots were established 1 year before measurements to minimize effects of disturbance (For further details see Dubbert et al. (2013)).

After a base line observation, all plots were watered with 20 mm water within one hour using watering cans. The water showed an oxygen isotopic signature of -139.5‰ to trace the influence of different vegetation components on water infiltration. All plots and the surrounding soil were watered equally to avoid lateral gradients and possible differences between trenched and control plots. Thereafter, all measurements were conducted in 7 diurnal cycles over the following 10-

12 days. The open and tree sites were watered independently, as the measurement setup did not allow highly resolved observations of all treatment plots at the same time. Environmental variables (PPFD; soil water content; vpd) were not significantly different between the first and second half of the observation period.

**2.2 Environmental variables and plant parameters**

5    Photosynthetic photon flux density (PPFD) was measured at both sites at approximately 1.5 m height (PPFD, LI-190SB, LI-COR, Lincoln, USA). Rainfall (ARG100 Rain gauge, Campbell Scientific, Logan, UT, USA), air temperature, and relative humidity (rH, CS-215 Temperature and Relative Humidity Probe, Campbell Scientific, Logan, UT, USA) were measured and 30 min averages were stored by a data logger (CR10x, Campbell Scientific, Logan, UT, USA). Soil temperature (custom built pt-100 elements)  was measured at -5 cm depth on vegetation and bare soil plots at both sites

10    and 60 min averages were stored in a data logger (CR1000, Campbell Scientific, Logan, UT, USA; 4 sensors per depth and treatment). Temperature at the soil surface was manually measured on each measurement day in diurnal cycles corresponding with the gas exchange measurements using temperature probes (GMH 2000, Greisinger electronic, Regenstauf, Germany). Volumetric soil water content ($\theta_s$, 10hs, Decagon, Washington, USA) was measured in -5, -15, -30 and -60 cm depth  on vegetation and bare soil plots at both sites and 60 min averages were stored in a data

15    logger (CR1000, Campbell Scientific, Logan, UT, USA; 4 sensors per depth and treatment).

Living aboveground biomass of herbaceous plants was determined destructively on five randomly selected, 40 × 40 cm plots at the beginning and end of the experiment in the open and under the trees. All green fresh aboveground plant biomass was collected, divided by species, dried (60 °C, 48 hours) and weighed. Below ground biomass was sampled with soil cores in -5, -15, -30, and -60 cm depth. Oven dried soil was sieved and root biomass was determined gravimetrically. 80 % of root

20    biomass was distributed between -5 to -15 cm depth. Only 5% was distributed above -5 cm and 15% between -20 to -35 cm depth. 
[revised manuscript text omitted]

 . However, (Dubbert et al. , 2014c) only observed precipitation events of light intensity during the period of interest. The present study reports on high intensity precipitation events. Furthermore, aboveground vegetation cover and biomass were reduced by 55 and 30 %, respectively, owing to the additional severe winter/spring drought in 2012. It is thus likely that such a drastic reduction in understory canopy cover eliminates much of the beneficial understory effects on the ecosystem water balance. This unexpected turn in effect direction with increasing precipitation intensity, which depends on vegetation cover and atmospheric evapotranspirative demand, potentially plays a strong role for the water balance of the ecosystem in the course of ongoing climate change scenarios since the occurrence of extreme precipitation events is expected to increase (IPCC, 2013).

Tree shading had a tremendous impact on the microclimate above understory plant and soil surfaces, but effects on infiltration amount could only be observed on vegetated plots. Reductions of the daily sum of global radiation $\sum R_g$ by 72% and daily peak soil temperatures $T_{S,5cm}$ up to 22% (Fig. 1) generated favorable conditions. Limited instantaneous evaporation from plant surfaces as described above led to 71% higher infiltration amounts (Fig. 3), whereas the  high infiltration amounts on bare soil plots were unaffected by tree shading. This confirms part one of hypothesis III on vegetated plots. Previous studies reported similar, positive feedbacks of tree cover for the hydrological cycle in savannah-type ecosystems related to shading effects (Eldridge and Freudenberger, 2005). Effects of altered soil hydraulic properties beneath tree crowns, like the amount of preferential flow fostering infiltration (Bargués Tobella et al., 2014) could not be identified in this study. Supporting findings are given by (Bhark and Small,  D'Odorico and Porporato, 2006). Considering the projected shading by crown cover of the tree layer (minimum of 30% at noon, increasing during the rest of the day, (Piayda et al. 2015), the infiltration enhancement has potentially large benefits on the ecosystem level. A former study of David et al. (2006) under comparable climatic and stand density conditions estimated only minor interception losses of 8% with respect to total canopy throughfall, due to low canopy cover typical for cork oak systems. However, the integral balance of canopy interception losses, increased infiltration and other benefits of tree cover (compare Joffre and Rambal (1993) and Dubbert et al. (2014c) in this ecosystem could not be analyzed in this study and needs further investigations with regard to tree density and age.

Subsurface distribution of soil water $\theta$ was systematically lower at depths below -20 cm at tree sites compared to open sites (Fig. 3). This clearly indicates the enhanced water extraction by tree roots, similar to results of Dubbert et al. (2014b). The observed pattern could not be changed by the event water pulse of 20 mm per hour, equal to a rain event of high intensity on this site. That explains the intense drought stress understory plants suffer during the transition

period from moist spring to dry summer, leading to earlier dieback under tree cover (Dubbert et al., 2014b; Moreno, 2008). and contradict part two of hypothesis III. The depth distribution of event water is very similar on bare soil plots that show an over all overall deeper infiltration of more water than the vegetated plots, caused by the higher infiltration amounts shown before. This shortcoming negative effect could partially be compensated by higher infiltration amounts below tree shading, but was consumed by tree water uptake from deeper depths within one day. During these dry conditions, pre-event water is located in small pores under high matrix potentials. Infiltrating event water partially displaced pre—event water downwards (Fig. 3) and additionally filled larger pores in the top soil. Thus, event water is more subject to evaporation due to lower matrix potentials in bigger pores than pre-event water. This observation is supported by a rapid decrease of event water content throughout the experiment.

**4.2 Dynamic responses of event water-use and plasticity of water uptake depth**

Successful biomass production of herbaceous vegetation highly depends on soil water availability in upper soil layers hosting the root system. Occasional precipitation events control the soil water regime (Porporato et al., 2004) which are prone to substantial changes in future climate change scenarios by stronger short term fluctuations of drought events (IPCC, 2013). Thus, a rapid adaptation of preferential root water uptake depth is crucial. This is particularly important for herbaceous vegetation in order to maximize the utilization of different soil water pools for a successful seed production, longevity and inter species competition (Ehleringer and Dawson, 1992; Rodriguez-Iturbe, 2000). It could be clearly shown that understory transpiration $T$ responded slower to an incoming precipitation pulse than soil evaporation $E$, with a time lag of about 24h. ET on vegetated plots and E on bare soil plots showed equally high peaks and a comparable decline until the end of the experiment, providing no evidence for higher water losses due to the presence of understory and contradicting part one of hypothesis I. During the entire experiment, $E$ was the dominant flux on both, tree and open sites, with a comparable contribution of transpiration $T$ to evapotranspiration $ET$ of 36% and 41% (Fig. 5), respectively. This small loss of productive transpiration water originates on one hand from the longer time response lag of $T$. On, on the other hand that from only little event water reaches reaching deeper soil layers, where understory plants have their main root water uptake depth prior to the precipitation pulse. Event water use of the understory vegetation was overall low, since no shift of root water uptake depth could be observed within the nine days of the experiments (Fig. 7) leading to comparably small isotopic depletion of bulk leaf water and transpiration (Fig. 4). 4), which supports hypothesis II. This is in agreement with previous findings where annual savannah species were not fast enough readjusting their water extraction depth in order to exploit precipitation water more efficiently (Asbjornsen et al., 2008; Kulmatiski and Beard, 2013). More importantly, during that period of the year the dry conditions in the upper soil layers forces understory plants in the direct vicinity of trees to compete for soil water at lower depths where the trees have their roots (i.e. tree sites). This observation clearly opposes the widely discussed two-layer hypothesis, proposing independent ecological niches for root water uptake of trees and understory plants in savannahs in order to avoid competition (Hipondoka et al., 2003; Holdo and Planque, 2013; Kulmatiski et al., 2010; Walter et al., 1971). Quite the contrary, previous findings of, e.g. Pryardarshini et al. (2015), suggest that tree based soil

water redistribution by hydraulic lift (Dawson, 1993) is an important contribution in water limited ecosystems like savannahs. This is a possible explanation for understory root water uptake at the depth of the first tree roots, as we found it in our study. Moreover, exponential soil profiles of plant available nitrogen causes a coupled water and nutrient competition between herbs and trees in this ecosystem during spring (Dubbert et al., 2014., 2014b). Modeling studies of e.g. Nippert et al. (2015) already suggested that understory plants do not exploit all accessible soil layers (including the top layers with high drought risk) in order to maximize water availability. Lower, but more resilient production is achieved instead by limiting root growth and water uptake to deeper depths, which could be confirmed by this study. It has to be additionally considered that the herbaceous vegetation already reached its growth peak when the experiment was conducted and thus maximizing root water uptake might not be a priority for the understory community past the growth peak and during seed production. Dubbert et al. (2014b) showed that the understory community is strongly adapted on a small spatial scale to the presence of oak trees regarding its species composition and overall vegetation period length. This is also observed in this study, with grasses dominating the understory community below the trees and forbs dominating in open areas. Effectively this leads to an earlier seed production and senescence of less drought tolerant grasses in water competition with trees and a longer vegetation period of drought tolerant native forbs (i.e. *Tuberaria guttata* or *Tolpis barbata*) in open areas. Consequently, while understory species in the open area remained a net sink for carbon during the entire experiment, the understory community below the trees was at the verge of senescence and turned into a net source for carbon at the last experimental date (Fig. A2), adding explanation to the site-specific differences of transpiration rate in response to event water (Fig. 5).

Recently, Volkmann et al. (2016a) used a similar flux / isotope approach to test the widespread dogma that plant water uptake depth is primarily controlled by root density distribution. While grassland species did not strongly alter their uptake pattern during the measurement campaign their water uptake depth profile was not in accordance with their root density distribution, with 85 % in the upper 10 cm of the soil profile. This clearly indicates that adapting the water uptake to soil water availability plays a role, but probably on longer time scales than what we observed during the 10 day's lasting experiment. Therefore, theThe development of membrane-based *in-situ* methods of soil water (Gaj et al., 2015; Rothfuss et al., 2015; Volkmann et al., 2016a) and, xylem sap sampling (Volkmann et al., 2016b) will advance the studies of dynamic changes in eco hydrological soil and transpiration (Dubbert et al., 2014a; Dubbert et al., 2017) will advance the studies of dynamic changes in ecohydrological soil-vegetation feedbacks in the future. Furthermore, the coupling of isotope laser spectroscopes to gas-exchange chambers and soil or xylem equilibration probes overcomes the cost and time consuming classical destructive sampling methods. Recent studies (i.e. (Orlowski et al., 2013)) showed significant isotopic deviations between actual soil water that is available for the plants and water that is cryogenically extracted from soil samples depending on soil the type. While we did not observe this in the sandy soils at our study site, these effects might severely hamper the usefulness of destructive soil sampling techniques in clay or loam soils. The newly developed in —situ techniques will thus facilitate cost—effective measurements of soil or xylem isotopic signatures with highest resolution,

enhancing our capacity to study the dynamics in soil water infiltration, in the uptake of water by plants and in the partitioning of evapotranspiration.

**5 Conclusion**

In this study, the various interactions between understory vegetation and trees of a Mediterranean cork oak woodland affecting the ecosystem water flows could be  quantified. The immediate on-site determination (with CRDS) of the isotope ratios from different soil and ecosystem compartments in combination with in situ sampling methods enhanced the resolution, precision and reliability of our results. This facilitated the tracing of the fate of rain pulse transitions through a typical Mediterranean savannah ecosystem using stable water isotopes.

Regardless of the presence of vegetation , the total evapotranspirative water loss of soil and understory remains unchanged, but infiltration rates decreased by 24% (hypothesis I rejected). Still, the amount of unproductive evaporation is largely reduced, in favour of transpiration. Adjustments of main root water uptake depth to changing soil water availability after rain pulses could not be observed (hypothesis II supported). Consequently the understory plants could not utilize the applied precipitation of 20 mm. Hence, these understory plants were forced into water competition with trees, rooting at deeper soil layers. Crown shading of cork oak trees altered micro climatic conditions, thus fostering infiltration and considerably reducing understory and soil evapotranspiration (hypothesis III, part one supported). Despite these benefits, understory plants in immediate vicinity of trees suffer from systematically lower soil moistures in deeper layers leading to premature senescence at the onset of drought (hypothesis III, part two rejected).

**Appendix A**

[Figure]

**Figure A1: Aboveground biomass on vegetated plots during the experiment time given for each genus. Standard errors are not given for the sake of clarity, but amount on average 30% of displayed genus biomass.**

[Figure]

**Figure A2: Mean midday net ecosystem exchange (NEE) of the understory vegetation at the open site (white circles) and the tree site (dark grey circles).**

**Author contribution**

Arndt Piayda and Maren Dubbert contributed equally to experimental work, data analysis and writing the manuscript. Rolf Siegwolf proofread the manuscript. Matthias Cuntz contributed to data analysis and proofread the manuscript. Christiane Werner contributed to field work and proofread the manuscript.

5 **Competing interests**

The authors declare that they have no conflict of interest.

**Acknowledgements**

We gratefully acknowledge excellent help in the laboratory by Ilse Thaufelder. Funding for this study was provided by the DFG grants to CW and MC (WATERFLUX Project: # WE 2681/6-1, # CU 173/2-1), as well as MD (# DU 1688/1-1).

10 **References**

[revised manuscript text omitted]

---

## Referee Report (RR1)

**Review**
Piayda and Dubbert et al., 2016 Quantification of dynamic soil – vegetation feedbacks following an isotopically labelled precipitation pulse, Biogiosciences Discuss., doi:10.5194/bg-2016-451

The authors made a commendable effort on improving their manuscript and I am pleased with the changes made having resulted in a high-quality research article. Some minor issues should still be addressed (see below). Thereafter I am happy to endorse this interesting work for publication in Biogeosciences.

***Minor Comments:***

a) The following points mainly regard implementation of requested changes in the manuscript being neglected and only addressed in the author's response:

*Previous communication:*

*Page 3 Line 16: Please expand on possible effects of meshes used for bare soil plots on water infiltration*

*Page 4 Line 6: fresh material was harvested, what was the proportion of already dry material, particularly in comparison to previous study of Dubbert et al. during a non-drought year, and the different effects of plant cover on infiltration reported in the discussion. This may have also reflected on the event water use in transpiration.*

*Page 5 Line 5: Leaf sampling did not affect ET in the vegetation plots? How big was the reduction of leaf area through sampling? Could this have affected the temporal progress of T from event water? Please elaborate on this here.*

Authors responded adequately. However, text on author's opinion on the effects of the meshes, proportion of dry material and leaf sampling on results being negligible should still be added to the manuscript, as this is information of importance to the reader.

b) Check usage of brackets with citations throughout the manuscript: E.g. Page 10 Line 24: Dubbert et al. (2014c) instead (Dubbert et al., 2014c)

c) Page 12 Line 28: I suggest citing Fig. A1 after "….in open areas"

---

## Author Response (AR2)

**Final author response to Anonymous Referee # 1 (RC1)**

*Referee comments* – Author response

**General Comments:**

*The authors made a commendable effort on improving their manuscript and I am pleased with the changes made having resulted in a high-quality research article. Some minor issues should still be addressed (see below). Thereafter I am happy to endorse this interesting work for publication in Biogeosciences.*

The authors are thankful for the general appreciation of the improvements made on the manuscript during the review and the recommendation for publication in *Biogeosciences* by Anonymous referee # 1.

**Minor Comments:**

*Page 3 Line 16: Please expand on possible effects of meshes used for bare soil plots on water infiltration*

The requested information in the manuscript was completed with the following information, previously only stated in the author's response: The sites were kept vegetation free just by regular weeding. We expect no influence of the mesh on infiltration, since the plots were installed one year before the experiment and processes like preferential flow along the mesh is unlikely.

*Page 4 Line 6: fresh material was harvested, what was the proportion of already dry material, particularly in comparison to previous study of Dubbert et al. during a non-drought year, and the different effects of plant cover on infiltration reported in the discussion. This may have also reflected on the event water use in transpiration.*

The requested information in the manuscript was completed with the following information, previously only stated in the author's response: Total aboveground biomass was relatively low compared to previous years between 42 and 78 g m-2 (see Fig. A1) with a minimal fraction of dry biomass due to the considerable winter/spring drought in the hydrological year 2012 (Costa e Silva et al., 2015; Dubbert et al., 2014b; Piayda et al., 2014). Dry biomass from the previous season was removed from the plots at the end of summer 2011.

*Page 5 Line 5: Leaf sampling did not affect ET in the vegetation plots? How big was the reduction of leaf area through sampling? Could this have affected the temporal progress of T from event water? Please elaborate on this here.*

The requested information in the manuscript was completed with the following information, previously only stated in the author's response: Mixed leaf samples of the herbaceous vegetation for water extraction were obtained in daily cycles in 2-hourly steps from 8:00 to 18:00 following a destructive sampling scheme affecting the overall amount of living biomass less than 5%. Thus, effects of destructive sampling on observed *ET* fluxes during the experiment are negligible.

*Check usage of brackets with citations throughout the manuscript*

All citations in the manuscript have been revised.

*Page 12 Line 28: I suggest citing Fig. A1 after "....in open areas"*

The figure reference was added.

[revised manuscript text omitted]

I. Presence of understory vegetation increases evapotranspirative water loss compared to bare soil, but foster infiltration due to shading.

II. Preferential root water uptake depth of understory plants is unaffected by changes in soil water availability after rain

10   pulses during drought.

III. Tree shading fosters infiltration of event water and reduces evapotranspiration generating favourable soil moisture conditions for understory plants.

**2 Material and methods**

**2.1 Study site and experimental design**

15   Measurements were conducted between May 26 and June 6 2012 in an open cork oak woodland (*Quercus suber* L.) in central Portugal, approximately 100 km north-east of Lisbon (N39°8'17.84'' W8°20'3.76''; Herdade de Machoqueira do Grou). The trees are widely spaced (209 individuals ha$^{-1}$) with a leaf area index of 1.1 and a gap probability of 0.7 (Piayda et al., 2015).

The herbaceous layer is dominated by native annual forbs and grasses. The site is characterized by Mediterranean climate,

20   with a 30 year long-term mean annual temperature of approximately 15.9 °C and annual precipitation of 680 mm (Instituto de Meteorologia, Lisbon). We established two sites: one directly under the oak crown projected area (tree site, ts) and another one in an adjacent open area (open site, os). Two types of plots (sized 40 × 80 cm) were installed in each site: bare soil plots with total exclusion of above and below-ground biomass (lateral root ingrowth was prevented by vertically inserted trenching meshes around the plots, mesh diameter < 1 µm, Plastok, Birkenhead, UK), and understory plots with herbaceous

25   vegetation (four plots per site and treatment). The sites were kept vegetation free just by regular weeding. We expect no influence of the mesh on infiltration, since the plots were installed one year before the experiment and processes like preferential flow along the mesh is unlikely All plots were established 1 year before measurements to minimize effects of disturbance (For further details see Dubbert et al. (2013)).

After a base line observation, all plots were watered with 20 mm water within one hour using watering cans. The water

30   showed an oxygen isotopic signature of -139.5‰ to trace the influence of different vegetation components on water infiltration. All plots and the surrounding soil were watered equally to avoid lateral gradients and possible differences between trenched and control plots. Thereafter, all measurements were conducted in 7 diurnal cycles over the following 10-

12 days. The open and tree sites were watered independently, as the measurement setup did not allow highly resolved observations of all treatment plots at the same time. Environmental variables (PPFD; soil water content; vpd) were not significantly different between the first and second half of the observation period.

**2.2 Environmental variables and plant parameters**

5    Photosynthetic photon flux density (PPFD) was measured at both sites at approximately 1.5 m height (PPFD, LI-190SB, LI-COR, Lincoln, USA). Rainfall (ARG100 Rain gauge, Campbell Scientific, Logan, UT, USA), air temperature, and relative humidity (rH, CS-215 Temperature and Relative Humidity Probe, Campbell Scientific, Logan, UT, USA) were measured and 30 min averages were stored by a data logger (CR10x, Campbell Scientific, Logan, UT, USA). Soil temperature (custom built pt-100 elements) was measured at -5 cm depth on vegetation and bare soil plots at both sites and 60 min averages were

10    stored in a data logger (CR1000, Campbell Scientific, Logan, UT, USA; 4 sensors per depth and treatment). Temperature at the soil surface was manually measured on each measurement day in diurnal cycles corresponding with the gas exchange measurements using temperature probes (GMH 2000, Greisinger electronic, Regenstauf, Germany). Volumetric soil water content ($\theta_s$, 10hs, Decagon, Washington, USA) was measured in -5, -15, -30 and -60 cm depth on vegetation and bare soil plots at both sites and 60 min averages were stored in a data logger (CR1000, Campbell Scientific, Logan, UT, USA; 4

15    sensors per depth and treatment).

Living aboveground biomass of herbaceous plants was determined destructively on five randomly selected, 40 × 40 cm plots at the beginning and end of the experiment in the open and under the trees. All green fresh aboveground plant biomass was collected, divided by species, dried (60 °C, 48 hours) and weighed. Below ground biomass was sampled with soil cores in -5, -15, -30, and -60 cm depth. Oven dried soil was sieved and root biomass was determined gravimetrically. 80 % of root

20    biomass was distributed between -5 to -15 cm depth. Only 5% was distributed above -5 cm and 15% between -20 to -35 cm depth. Total aboveground biomass was relatively low compared to previous years between 42 and 78 g m-2 (see Fig. A1) with a minimal fraction of dry biomass, due to the considerable winter/spring drought in the hydrological year 2012 (Costa e Silva et al., 2015; Dubbert et al., 2014b; Piayda et al., 2014). Dry biomass from the previous season was removed from the plots at the end of summer 2011. While total aboveground biomass was similar between plots, species composition and

25    relative dominance differed with the open sites being dominated by *Tuberaria guttata* and the tree sites by grass and legume species (Dubbert et al., 2014b).

**2.3 Cavity Ring-Down Spectrometer based gas-exchange flux and δ$^{18}$O measurements**

Water fluxes and isotopic composition were measured with a Wavelength Scanned Cavity Ring-Down Spectrometer (WS-CRDS, Picarro, Santa Clara, USA) in combination with custom built soil chambers (following the design of Pape et al.

30    (2009)) in an open gas exchange system (n=3 per treatment and experimental site). Background and sampling air were measured alternately after stable values were reached. A five minutes interval average was used for the calculation of evapotranspiration (*ET*) and evaporation (*E*). *ET* and *E* were calculated according to von Caemmerer and Farquhar (1981).

Oxygen isotope compositions of soil evaporation (bare soil plots) as well as evapotranspiration of the understory (vegetation plots) were estimated using a mass balance approach (Dubbert et al., 2013; Dubbert et al., 2014c):

$$\delta_E = \frac{u_{out}w_{out}\delta_{out} - u_{in}w_{in}\delta_{in}}{u_{out}w_{out} - u_{in}w_{in}} \tag{1},$$

$$\delta_E = \frac{w_{out}\delta_{out} - w_{in}\delta_{in}}{w_{out} - w_{in}} - \frac{w_{in}w_{out}(\delta_{out} - \delta_{in})}{w_{out} - w_{in}} \tag{2},$$

where $u$ is the flow rate [mol(air) s$^{-1}$], $w$ is the mole fraction [mol(H$_2$O) mol(air)$^{-1}$] and $\delta$ is isotope value of the incoming (*in*) and outgoing (*out*) air stream of the chamber. Flow rates are measured with humid air so that conservation of dry air gives $u_{in}(1-w_{in}) = u_{out}(1-w_{out})$, which leads to Eq. (2). The second term in Eq. (2) corrects for the increased gas flow in the chamber due to addition of water by transpiration. In addition to isotopic signatures of soil evaporation and understory evapotranspiration, the oxygen isotope signatures of ambient water vapor (in 9 m height) were measured with the CRDS. All measurements were conducted as diurnal courses with 5-6 measurement points between 7 a.m. and 7 p.m. For more details about the chamber design and measurement setup see Dubbert et al. (2013).

**2.4 Sampling and measurement of $\delta^{18}$O of soil and leaf water**

Soil samples for water extraction and $\delta^{18}O$ analysis were taken on vegetated and bare soil plots using a soil corer. Samples were collected from the soil surface (0-0.5 cm depth), -2, -5, -10, -15, -20, and -40 cm soil depths (n=4 per depth and treatment) usually during midday, but on the day of irrigation directly proceeding the irrigation pulse and additionally at 18:00. Mixed leaf samples of the herbaceous vegetation for water extraction were obtained in daily cycles in 2-hourly steps from 8:00 to 18:00, following a destructive sampling scheme affecting the overall amount of living biomass less than 5%. Thus, effects of destructive sampling on observed *ET* fluxes during the experiment are negligible. 
[revised manuscript text omitted]
. 5). This is in contrast to previous studies, which reported beneficial effects of plant cover on daily sums of infiltration during the same period at the onset of drought in 2011 (Dubbert et al., 2014c). However, Dubbert et al. (2014c) only observed precipitation events of light intensity during the period of interest. The present study reports on high intensity precipitation events. Furthermore, aboveground vegetation cover and biomass were reduced by 55 and 30 %, respectively, owing to the additional severe winter/spring drought in 2012. It is thus likely that such a drastic reduction in understory canopy cover eliminates much of the beneficial understory effects on the ecosystem water balance. This unexpected turn in effect direction with increasing precipitation intensity, which depends on vegetation cover and atmospheric evapotranspirative demand, potentially plays a

strong role for the water balance of the ecosystem in the course of ongoing climate change scenarios since the occurrence of extreme precipitation events is expected to increase (IPCC, 2013).

Tree shading had a tremendous impact on the microclimate above understory plant and soil surfaces, but effects on infiltration amount could only be observed on vegetated plots. Reductions of the daily sum of global radiation $\sum R_g$ by 72% and daily peak soil temperatures $T_{S,5cm}$ up to 22% (Fig. 1) generated favorable conditions. Limited instantaneous evaporation from plant surfaces as described above led to 71% higher infiltration amounts (Fig. 3), whereas the high infiltration amounts on bare soil plots were unaffected by tree shading. This confirms part one of hypothesis III on vegetated plots. Previous studies reported similar, positive feedbacks of tree cover for the hydrological cycle in savannah-type ecosystems related to shading effects (Eldridge and Freudenberger, 2005). Effects of altered soil hydraulic properties beneath tree crowns, like the amount of preferential flow fostering infiltration (Bargués Tobella et al., 2014) could not be identified in this study. Supporting findings are given by Bhark and Small (2003); D'Odorico and Porporato (2006). Considering the projected shading by crown cover of the tree layer (minimum of 30% at noon, increasing during the rest of the day, (Piayda et al., 2015)), the infiltration enhancement has potentially large benefits on the ecosystem level. A former study of David et al. (2006) under comparable climatic and stand density conditions estimated only minor interception losses of 8% with respect to total canopy throughfall, due to low canopy cover typical for cork oak systems. However, the integral balance of canopy interception losses, increased infiltration and other benefits of tree cover (compare Joffre and Rambal (1993) and Dubbert et al. (2014c)) in this ecosystem could not be analyzed in this study and needs further investigations with regard to tree density and age.

Subsurface distribution of soil water $\theta$ was systematically lower at depths below -20 cm at tree sites compared to open sites (Fig. 3). This clearly indicates the enhanced water extraction by tree roots, similar to results of Dubbert et al. (2014b). The observed pattern could not be changed by the event water pulse of 20 mm per hour, equal to a rain event of high intensity on this site. That explains the intense drought stress understory plants suffer during the transition period from moist spring to dry summer, leading to earlier dieback under tree cover (Dubbert et al., 2014b; Moreno, 2008) and contradict part two of hypothesis III. The depth distribution of event water is very similar on bare soil plots that show an overall deeper infiltration of more water than the vegetated plots, caused by the higher infiltration amounts shown before. This negative effect could partially be compensated by higher infiltration amounts below tree shading, but was consumed by tree water uptake from deeper depths within one day. During these dry conditions, pre-event water is located in small pores under high matrix potentials. Infiltrating event water partially displaced pre-event water downwards (Fig. 3) and additionally filled larger pores in the top soil. Thus, event water is more subject to evaporation due to lower matrix potentials in bigger pores than pre-event water. This observation is supported by a rapid decrease of event water content throughout the experiment.

**4.2 Dynamic responses of event water-use and plasticity of water uptake depth**

Successful biomass production of herbaceous vegetation highly depends on soil water availability in upper soil layers hosting the root system. Occasional precipitation events control the soil water regime (Porporato et al., 2004) which are

prone to substantial changes in future climate change scenarios by stronger short term fluctuations of drought events (IPCC, 2013). Thus, a rapid adaptation of preferential root water uptake depth is crucial. This is particularly important for herbaceous vegetation in order to maximize the utilization of different soil water pools for a successful seed production, longevity and inter species competition (Ehleringer and Dawson, 1992; Rodriguez-Iturbe, 2000). It could be clearly shown

5    that understory transpiration $T$ responded slower to an incoming precipitation pulse than soil evaporation $E$, with a time lag of about 24h. $ET$ on vegetated plots and $E$ on bare soil plots showed equally high peaks and a comparable decline until the end of the experiment, providing no evidence for higher water losses due to the presence of understory and contradicting part one of hypothesis I. During the entire experiment, $E$ was the dominant flux on both, tree and open sites, with a comparable contribution of transpiration $T$ to evapotranspiration $ET$ of 36% and 41% (Fig. 5), respectively. This small loss of

10   transpiration water originates on one hand from the longer time response lag of $T$, on the other hand from only little event water reaching deeper soil layers, where understory plants have their main root water uptake depth. Event water use of the understory vegetation was overall low, since no shift of root water uptake depth could be observed within the nine days of the experiments (Fig. 7) leading to comparably small isotopic depletion of bulk leaf water and transpiration (Fig. 4), which supports hypothesis II. This is in agreement with previous findings where annual savannah species were not fast enough

15   readjusting their water extraction depth in order to exploit precipitation water more efficiently (Asbjornsen et al., 2008; Kulmatiski and Beard, 2013). More importantly, during that period of the year the dry conditions in the upper soil layers forces understory plants in the direct vicinity of trees to compete for soil water at lower depths where the trees have their roots (i.e. tree sites). This observation clearly opposes the widely discussed two-layer hypothesis, proposing independent ecological niches for root water uptake of trees and understory plants in savannahs in order to avoid competition (Hipondoka

20   et al., 2003; Holdo and Planque, 2013; Kulmatiski et al., 2010; Walter et al., 1971). Moreover, exponential soil profiles of plant available nitrogen causes a coupled water and nutrient competition between herbs and trees in this ecosystem during spring (Dubbert et al., 2014b). Modeling studies of e.g. Nippert et al. (2015) already suggested that understory plants do not exploit all accessible soil layers (including the top layers with high drought risk) in order to maximize water availability. Lower, but more resilient production is achieved instead by limiting root growth and water uptake to deeper depths, which

25   could be confirmed by this study. It has to be additionally considered that the herbaceous vegetation already reached its growth peak when the experiment was conducted and thus maximizing root water uptake might not be a priority for the understory community past the growth peak and during seed production. Dubbert et al. (2014b) showed that the understory community is strongly adapted on a small spatial scale to the presence of oak trees regarding its species composition and overall vegetation period length. This is also observed in this study, with grasses dominating the understory community

30   below the trees and forbs dominating in open areas (Fig. A1). Effectively this leads to an earlier seed production and senescence of less drought tolerant grasses in water competition with trees and a longer vegetation period of drought tolerant native forbs (i.e. *Tuberaria guttata* or *Tolpis barbata*) in open areas. Consequently, while understory species in the open area remained a net sink for carbon during the entire experiment, the understory community below the trees was at the verge of

senescence and turned into a net source for carbon at the last experimental date (Fig. A2), adding explanation to the site-specific differences of transpiration rate in response to event water (Fig. 5).

Recently, Volkmann et al. (2016a) used a similar flux / isotope approach to test the widespread dogma that plant water uptake depth is primarily controlled by root density distribution. While grassland species did not strongly alter their uptake pattern during the measurement campaign their water uptake depth profile was not in accordance with their root density distribution, with 85 % in the upper 10 cm of the soil profile. This clearly indicates that adapting the water uptake to soil water availability plays a role, but probably on longer time scales than what we observed during the 10 day's lasting experiment. The development of membrane-based *in-situ* methods of soil water (Gaj et al., 2015; Rothfuss et al., 2015; Volkmann et al., 2016a), xylem sap sampling (Volkmann et al., 2016b) and transpiration (Dubbert et al., 2014a; Dubbert et al., 2017) will advance the studies of dynamic changes in ecohydrological soil-vegetation feedbacks in the future. Furthermore, the coupling of isotope laser spectroscopes to gas-exchange chambers and soil or xylem equilibration probes overcomes the cost and time consuming classical destructive sampling methods. Recent studies (Orlowski et al., 2013) showed significant isotopic deviations between actual soil water that is available for the plants and water that is cryogenically extracted from soil samples depending on soil type. While we did not observe this in sandy soils at our study site, these effects might severely hamper the usefulness of destructive soil sampling techniques in clay or loam soils. The newly developed in situ techniques will thus facilitate cost-effective measurements of soil or xylem isotopic signatures with highest resolution, enhancing our capacity to study the dynamics in soil water infiltration, in the uptake of water by plants and in the partitioning of evapotranspiration.

**5 Conclusion**

In this study, the various interactions between understory vegetation and trees of a Mediterranean cork oak woodland affecting the ecosystem water flows could be quantified. The immediate on-site determination (with CRDS) of the isotope ratios from different soil and ecosystem compartments in combination with in situ sampling methods enhanced the resolution, precision and reliability of our results. This facilitated the tracing of the fate of rain pulse transitions through a typical Mediterranean savannah ecosystem using stable water isotopes.

Regardless of the presence of vegetation, the total evapotranspirative water loss of soil and understory remains unchanged, but infiltration rates decreased by 24% (hypothesis I rejected). Still, the amount of unproductive evaporation is largely reduced, in favour of transpiration. Adjustments of main root water uptake depth to changing soil water availability after rain pulses could not be observed (hypothesis II supported). Consequently the understory plants could not utilize the applied precipitation of 20 mm. Hence, these understory plants were forced into water competition with trees, rooting at deeper soil layers. Crown shading of cork oak trees altered micro climatic conditions, thus fostering infiltration and considerably reducing understory and soil evapotranspiration (hypothesis III, part one supported). Despite these benefits, understory

plants in immediate vicinity of trees suffer from systematically lower soil moistures in deeper layers leading to premature senescence at the onset of drought (hypothesis III, part two rejected).

**Appendix A**

[Figure]

5     **Figure A1: Aboveground biomass on vegetated plots during the experiment time given for each genus. Standard errors are not given for the sake of clarity, but amount on average 30% of displayed genus biomass.**

[Figure]

**Figure A2: Mean midday net ecosystem exchange (NEE) of the understory vegetation at the open site (white circles) and the tree site (dark grey circles).**

**Author contribution**

Arndt Piayda and Maren Dubbert contributed equally to experimental work, data analysis and writing the manuscript. Rolf Siegwolf proofread the manuscript. Matthias Cuntz contributed to data analysis and proofread the manuscript. Christiane Werner proofread the manuscript.

**Competing interests**

The authors declare that they have no conflict of interest.

**Acknowledgements**

We gratefully acknowledge excellent help in the laboratory by Ilse Thaufelder. Funding for this study was provided by the DFG grants to CW and MC (WATERFLUX Project: # WE 2681/6-1, # CU 173/2-1), as well as MD (# DU 1688/1-1).

**References**

[revised manuscript text omitted]